# Outward Odyssey: Improving Reward Models with Proximal Policy Exploration for Preference-Based Reinforcement Learning

## Abstract

Reinforcement learning (RL) heavily depends on well-designed reward functions, which can be challenging to create and may introduce biases, especially for complex behaviors. Preference-based RL (PbRL) addresses this by using preference feedback to construct a reward model that reflects human preferences, yet requiring considerable human involvement. To alleviate this, several PbRL methods aim to select queries that need minimal feedback. However, these methods do not directly enhance the data coverage within the preference buffer. In this paper, to emphasize the critical role of preference buffer coverage in determining the quality of the reward model, we first investigate and find that a reward model's evaluative accuracy is the highest for trajectories within the preference buffer's distribution and significantly decreases for out-of-distribution trajectories. Against this phenomenon, we introduce the **Proximal Policy Exploration (PPE)** algorithm, which consists of a *proximal-policy extension* method and a *mixture distribution query* method. To achieve higher preference buffer coverage, the *proximal-policy extension* method encourages active exploration of data within near-policy regions that fall outside the preference buffer's distribution. To balance the inclusion of in-distribution and out-of-distribution data, the *mixture distribution query* method proactively selects a mix of data from both outside and within the preference buffer's distribution for querying. PPE not only expands the preference buffer's coverage but also ensures the reward model's evaluative capability for in-distribution data. Our comprehensive experiments demonstrate that PPE achieves significant improvement in both preference feedback efficiency and RL sample efficiency, underscoring the importance of preference buffer coverage in PbRL tasks.

## 1 Introduction

In reinforcement learning (RL), the reward function is pivotal as it specifies the learning objectives and guides agents toward desired behaviors. Traditional RL has made significant achievements in complex domains such as gaming and robotics, largely due to the use of well-designed reward functions (Mnih et al., 2015; Silver et al., 2017; Degrave et al., 2022). Yet, constructing these functions presents significant challenges. The intricate process of designing suitable reward functions that accurately encapsulate complex behaviors like cooking or summarizing books is both time-consuming and prone to human cognitive biases (Wu et al., 2021; Hadfield-Menell et al., 2017; Abel et al., 2021; Li et al., 2023; Sorg, 2011). Additionally, embedding social norms into these functions remains an unresolved issue (Amodei et al., 2016).

An emerging alternative that addresses some of these challenges is preference-based reinforcement learning (PbRL). This approach bypasses the need for meticulously engineered rewards by leveraging overseer preferences between pairs of agent behaviors, which is typically fathered from human (Christiano et al., 2017; Ibarz et al., 2018; Lee et al., 2021b;a; Park et al., 2022; Liang et al., 2022; Shin et al., 2023; Tien et al., 2022). In PbRL, agents learn to optimize behaviors that align with the demonstrated human preferences, offering a more intuitive and flexible method for performing desired behaviors.

Despite its advantages, PbRL typically requires extensive preference feedback, which can be labor-intensive, time-consuming and sometimes infeasible to gather, potentially limiting its applicability in real-world settings where rapid adaptation is essential (Lee et al., 2021a; Park et al., 2022; Liang et al., 2022). To overcome these challenges, prior research has explored various strategies for improving feedback efficiency. These strategies include selecting the most informative queries to improve the quality of the learned reward function while minimizing the required teacher input (Lee et al., 2021b; Biyik & Sadigh, 2018; Sadigh et al., 2017; Bıyık et al., 2020). Also, techniques such as sampling based on ensemble disagreements, mutual information, or behavior entropy have been employed to target behaviors to refine the overall reward model more effectively (Christiano et al., 2017; Lee et al., 2021a; Shin et al., 2023; Biyik & Sadigh, 2018; Bıyık et al., 2020). Moreover, QPA (Hu et al., 2023) ensures that both queries and policy learning progress concurrently, significantly reducing feedback unrelated to the current policy, thereby enhancing feedback efficiency. However, these methods overlook the investigation of the relationship between the preference buffer and the effectiveness of the reward model. This oversight can lead the reward model to inaccurately evaluate data that is out of the preference buffer's distribution, potentially leading to misguided policy improvements.

To address this issue, we focus on enhancing the coverage of the preference buffer. Basically, our findings revealed that the learned reward model provides more precise evaluations for trajectories that fall within the preference buffer's distribution. This insight led us to develop the Proximal Policy Exploration (PPE) algorithm. Firstly, we need to train an out-of-distribution (OOD) detection mechanism to evaluate whether newly encountered data from the environment falls outside the preference buffer's distribution. Using the OOD degree measurement of the current data, we employ the *proximal-policy extension* method, which encourages the agent to explore data that, while beyond the preference buffer's distribution, still aligns closely with the current policy. Furthermore, we have designed the *mixture distribution query* method, which not only actively queries data outside the preference buffer's distribution but also queries a portion of the in-distribution data. The aim of this approach is to actively expand the preference buffer's coverage while avoiding a reduction in the reward model's evaluation accuracy for in-distribution trajectories due to insufficient volume of in-distribution data. By integrating these two methods, we are able to broaden the preference buffer's coverage and bolster the reliability of the reward model's evaluations for the near-policy distribution.

In summary, our contributions are threefold:

1. We introduce an OOD detection mechanism to ascertain whether data falls outside the preference buffer's distribution, and formulate the behavior policy resolution as a constrained optimization problem for exploring such data.

2. For this constrained optimization problem, we provide a closed-form approximation. Through this, we introduce the *proximal-policy extension* method in PPE, an analytical behavior policy that directly explores data outside the preference buffer's distribution. This approach actively enhances the coverage of the preference buffer.

3. We have found that the reliability of the reward model is heavily dependent on the data distribution; the reward model can only provide reliable assessments when there is sufficient data within the evaluated distribution. To address this, we propose a *mixture distribution query* method in PPE, which balances the volume of in-distribution and out-of-distribution query data, ensuring accurate evaluations by the reward model across different regions.

## 2 PRELIMINARIES

**Preference-based RL**   In PbRL, we consider an agent that interacts with an environment in discrete time steps. At each time step $t$, the agent at state $s_t$ selects an action $a_t$ based on its policy. Unlike traditional RL, where the environment returns a reward $r(s_t, a_t)$ evaluating the agent's behavior, PbRL employs preference feedback. Here, a teacher provides preferences between pairs of agent behaviors, which the agent uses to learn proxy rewards that align with human preferences, guiding the agent to adjust its policy (Christiano et al., 2017; Ibarz et al., 2018; Lee et al., 2021b; Sutton, 2018; Leike et al., 2018).

Formally, a behavior segment $\tau$ consists of a sequence of time-indexed observations and actions $\{(s_t, a_t), \ldots, (s_{t+H}, a_{t+H})\}$. Given a pair of segments $(\tau^0, \tau^1)$, the teacher gives their preference feedback signal $y_p$ among these segments, identifying preferred behaviors or marking segments as equally preferred or incomparable. The primary objective in PbRL is to train the agent to perform behaviors aligned with human with minimal feedback.

The PbRL learning process involves two main steps: (1) *Agent Learning*: The agent interacts with the environment to collect experiences and updates its policy using existing RL algorithms to maximize the sum of proxy rewards. (2) *Reward Learning*: The reward model $\hat{r}_\psi$ is optimized based on feedback received from the teacher, denoted as $(\tau^0, \tau^1, y_p) \sim \mathcal{D}_p$. This cyclical process continually refines both the policy and the reward model, detailed in Appendix A.

**OOD Detection**    Neural networks are known for making confident predictions, even when encountering out-of-distribution (OOD) samples (Nguyen et al., 2015; Goodfellow et al., 2014; Lakshminarayanan et al., 2017). A common approach for OOD detection involves fitting a generative model to the dataset, which assigns high probability to in-distribution samples and low probability to OOD ones. Although effective for simple, unimodal data, these methods can become computationally intensive when dealing with more complex and multimodal data. An alternative approach trains classifiers to act as more sophisticated OOD detectors (Lee et al., 2018).

In this study, we focus on Morse neural networks (Dherin et al., 2023), which train a generative model to produce an unnormalized density that equals to 1 at the dataset modes. We utilize this model to generate a metric that assesses the extent to which current data deviates from the preference buffer distribution. A Morse neural network produces an unnormalized density $M(x) \in [0, 1]$ on an embedding space $\mathbb{R}^e$, attaining a value of 1 at mode submanifolds and decreasing towards 0 when moving away from the mode (Dherin et al., 2023). The rate at which the value decreases is controlled by a Morse Kernel. More details about the Morse neural network can be found in Appendix B.

## 3    METHOD

In this chapter, we delve into the importance of preference buffer coverage for the reward model in our study and discuss strategies to actively expand this coverage.

### 3.1    WHY COVERAGE IS IMPORTANT? — A MOTIVATING EXAMPLE

We designed an experiment to observe the relationship between the effectiveness of the reward model and the coverage of transitions in the preference buffer used to train the reward model.

As shown in Figure 1, we set up an environment in a grid world where the robot can move in four directions: up, down, left, and right. Each cell in the grid world has an associated ground truth reward, which corresponds to a ground truth return for the robot's trajectory. It should be noted that Figure 1a serves as a schematic representation; in reality, the grid world is structured as a 9x9 grid. Additionally, the horizontal axes in Figures 1b and 1c represent the side lengths of the respective region, while the horizontal axis in Figure 1d represents the number of feedbacks.

We further designated two areas within the grid world as the training region and the evaluation region, as illustrated in Figure 1a . First, we uniformly sampled 1,000 trajectory pairs of length 3 in the training region. Based on the relative sizes of their ground truth returns, we assigned preference labels to these trajectory pairs and stored them in a preference buffer. Next, we trained a reward model using the data from the preference buffer with a Bradley-Terry loss. Finally, we evaluated all trajectories of length 6 in the evaluation region using the learned reward model to determine their merit. The correlation between the proxy returns computed by the reward model and the ground truth returns was assessed using the Spearman correlation coefficient to further analyze the effectiveness of the reward model.

Results displayed in Figure 1c indicate that a larger training region enhances the ability of the reward model, learned from the corresponding preference buffer, to effectively evaluate the merits of trajectories. This phenomenon is intuitive yet underscores the critical importance of increasing the coverage of the preference buffer over the transition space. Consider the policy optimization process: if the preference buffer does not comprehensively cover the transition distribution associated

with the current policy, the proxy rewards generated by the reward model may be unreliable, rendering the direction of policy optimization meaningless. Only with extensive coverage of the preference buffer can the reward model learned from it reliably evaluate a broader area. Based on this insight, it is essential to include the coverage of the preference buffer as an optimization objective within the pipeline of PbRL. Figure 1b demonstrates that the variance in outputs from ensemble reward models, given the same transition input, does not enable distinction of whether the transition belongs to the training region. Therefore, RUNE, proposed by Liang et al. (2022) cannot actively expand the preference buffer's coverage. Figure 1d shows that with the same training region, the more feedback used, the higher the evaluation accuracy of the trained reward model. This indicates that we cannot solely focus on exploring and collecting data outside the preference buffer distribution. It is also necessary to ensure that the new queries include a sufficient amount of in-distribution data. This balance is crucial to prevent the reward model from inaccurately evaluating regions it has already explored.

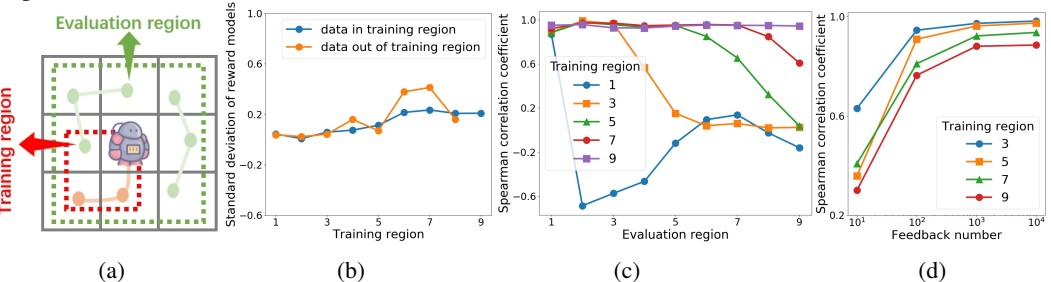

(a)        (b)        (c)        (d)

Figure 1: Observe the reward model's effectiveness in a random walk task with a grid world. **(a)**. Training the reward model with preference data generated from trajectory pairs within the training region marked by the red frame, and assessing the correlation between the proxy and ground truth returns across all trajectories in the evaluation region denoted by the green frame; **(b)**. The variance in the proxy rewards associated with transitions inside and outside of the training region changes in the size of the training region; **(c)**. The Spearman correlation coefficient between proxy returns and ground truth returns for all trajectories in various evaluation regions, using reward models trained with preference data from different training regions; **(d)**. The Spearman correlation coefficient varies with the number of feedbacks used to train the reward model in different training regions.

Consequently, to train a reliable reward model, it is essential not only for the agent to actively explore OOD data to expand the preference buffer coverage but also to ensure that there is a sufficient amount of in-distribution data within the preference buffer.

### 3.2 How to Expand Coverage of Preference Buffer? — Proximal Policy Exploration

Based on the observations in Section 3.1, we propose the PPE algorithm, which includes two core modules: the *proximal-policy extension* method to enhance preference buffer coverage, and the *mixture distribution query* method to balance the inclusion of in-distribution and out-of-distribution data. By leveraging transition uncertainty estimation, PPE combines these methods to develop a more reliable reward model within the current policy distribution.

**Leveraging Morse Neural Network for Transition Uncertainty Estimation**    Drawing inspiration from the work of Srinivasan & Knottenbelt (2024), we propose $f_\phi$ as a perturbation model that generates an action $\hat{a} = f_\phi(s, a)$. This implies that $\hat{a} = a$ only when the pair $(s, a)$ originates from the preference buffer $\mathcal{D}^p$. Simultaneously, the preference buffer $\mathcal{D}^p$ is composed of tuples $(\tau^0, \tau^1, y_p)$, where each segment $\tau$ is a sequence of state-action pairs $\{(s_t, a_t), \ldots, (s_{t+H}, a_{t+H})\}$. Based on this, we design the Morse Neural Network such that $M_\phi(s_i, a_j) = 0$ is valid only when $\{s_i, a_j\} \in \mathcal{D}^p$. In particular, we utilize a Radial Basis Function (RBF) kernel (Seeger, 2004) to shape the Morse Network, as illustrated in Eq.(1).

$$M_\phi(s, a) = 1 - K_{RBF}(f_\phi(s, a), a), \text{ where } K_{RBF}(z_1, z_2) = e^{-\frac{\lambda^2}{2}\|z_1 - z_2\|^2}. \tag{1}$$

Subsequently, we optimize this Morse Neural Network by minimizing the KL divergence between unnormalized measures (Amari, 2016), as detailed in Dherin et al. (2023). This can be expressed

as $D_{KL}(\mathcal{D}^p(s,a)\|1-M_\phi(s,a))$. Hence, in terms of $\phi$, this implies minimizing the loss depicted in Eq.(2). Additional details can be found in Appendix C.

$$L(\phi) = \frac{1}{N} \sum_{s,a \sim \mathcal{D}^p} \left[ \frac{\lambda^2}{2} \|f_\phi(s,a) - a\|^2 + \frac{1}{M} \sum_{a_u \sim \text{Uniform}(\mathcal{A})} \exp^{-\frac{\lambda^2}{2}\|f_\phi(s,a_u)-a_u\|^2} \right] \quad (2)$$

Here, $a_u$ signifies an action sampled from a uniform distribution over the corresponding action space, denoted as $\text{Uniform}(\mathcal{A})$. Furthermore, $M$ represents the number of samples drawn from $\text{Uniform}(\mathcal{A})$, while $N$ refers to the number of sampled $(s,a)$ pairs from $\mathcal{D}^p$. The parameter $\lambda$ is used to control the sensitivity of the Morse Neural Network to OOD transitions.

**Expanding Preference Buffer Coverage via Proximal-Policy Extension Method**   Observations from Figure 1c suggest that expanding the coverage of the preference buffer can enhance the ability of the trained reward model in evaluating the quality of trajectories. Particularly during the RL training process, Only when the trained reward model has a strong ability to evaluate the quality of trajectories within the proximal policy distribution can the risk of misguidance in policy improvement be reduced. Therefore, expanding the coverage of the preference buffer for the proximal policy distribution can further optimize policy improvement in PbRL.

Drawing on this insight, we have designed the *proximal-policy extension* method, to actively encourage the agent to explore data that falls outside the preference buffer distribution but within the vicinity of the current policy's distribution. The behavior policy $\pi_E$ used for exploration, is designed such that the state-action pairs $(s,a)$ it generates when interacting with the environment can support the distribution produced by the current target policy $\pi_T$. Formally, the behavior policy $\pi_E = \mathcal{N}(\mu_E, \Sigma_E)$ is defined as the solution to the constrained optimization problem in Eq.(3).

$$\max_{\mu,\Sigma} \mathbb{E}_{a \sim \mathcal{N}(\mu,\Sigma)} [M_\phi(s,a)],$$
$$\text{s.t. } D_{KL}(\mathcal{N}(\mu,\Sigma)|\mathcal{N}(\mu_T, \Sigma_T)) \le \epsilon. \quad (3)$$

Since we need to calculate the constrained optimization problem described in Eq.(3) in each interaction process, using readily available solvers would result in a significant consumption of computational resources. Therefore, we tighten the constraint conditions to obtain a closed-form approximate solution as shown in Proposition 1. This approach greatly reduces the computational cost of solving the constrained optimization problem, while achieving our desired objective of encouraging exploration of data out of the preference buffer distribution near the current policy distribution. The detailed derivation is presented in Appendix D.

**Proposition 1** *The behavior policy for exploration resulting from Eq.(3) has the form* $\pi_E = \mathcal{N}(\mu_E, \Sigma_E)$, *where*

$$\mu_E = \mu_T + \frac{\sqrt{2\epsilon} \cdot \Sigma_T [\nabla_a M_\phi(s,a)]_{a=\mu_T}}{\sqrt{[\nabla_a M_\phi(s,a)]_{a=\mu_T}^T \Sigma_T [\nabla_a M_\phi(s,a)]_{a=\mu_T}}}, and \ \Sigma_E = \Sigma_T. \quad (4)$$

**Mixture Distribution Query Selection**   In the previous section, we introduced an exploration method that enables the agent to explore a broader range of transitions that are out of the preference buffer but near the current policy distribution. These newly discovered transitions are stored in the replay buffer. Therefore, it becomes essential to have a query selection method that can select those segments that are out of the preference buffer and store them in the preference buffer.

Additionally, as inspired by the phenomenon demonstrated in Figure 1d, if we merely select those segments outside the preference buffer's distribution and store them in the preference buffer, it implies that the volume of data in the in-distribution region will not undergo substantial expansion. As a result, the evaluation capability of the trained reward model in the in-distribution region may become less reliable due to the lack of sufficient data in this area.

Taking all these factors into account, we propose the *mixture distribution query* method. This method aims to actively select out-of-distribution data to increase the preference buffer coverage, while also selecting some in-distribution data for query. This method not only proactively increases the

coverage of the preference buffer but also boosts the volume of in-distribution data, thereby ensuring the evaluation capability of the reward model in the in-distribution region.

Specifically, for all $\tau \in \mathcal{D}^{cp}$, we can express the degree of a segment of trajectory $\tau$ being out of the preference buffer distribution according to Eq.(5), where $\mathcal{D}^{cp}$ represents the data to be queried. A higher value indicates that the data is more likely to be in-distribution.

$$M_\phi(\tau) = \frac{1}{|\tau|} \sum_{(s,a) \in \tau} M_\phi(s, a) \tag{5}$$

The size of $\mathcal{D}^{cp}$ is not large, typically $|\mathcal{D}^{cp}| \ll |\mathcal{D}|$, especially when combined with the *policy-aligned query* technique proposed in QPA (Hu et al., 2023), the quantity of $\mathcal{D}^{cp}$ is further reduced. Under this premise, we can redistribute the sampling probability for $\tau \in \mathcal{D}^{cp}$.

As shown in Eq.(6), we designed two probability density functions $P^{in}(\cdot)$ and $P^{out}(\cdot)$ according to the degree of in-distribution and out-of-distribution, respectively representing the probability of sampling $\tau$ according to the degree of in-distribution and out-of-distribution. We use a mixture ratio $\kappa \in [0, 1]$ to control the proportion of samples drawn from each distribution. A larger $\kappa$ indicates a higher proportion of samples are drawn from $P^{out}(\cdot)$.

$$P^{in}(\tau) = \frac{1 - M_\phi(\tau)}{\sum_{\tau' \in \mathcal{D}^{cp}} [1 - M_\phi(\tau')]}$$
$$P^{out}(\tau) = \frac{M_\phi(\tau)}{\sum_{\tau' \in \mathcal{D}^{cp}} M_\phi(\tau')} \tag{6}$$

It's worth noting that, as mentioned in the preceding paragraph, we need to calculate $M_\phi(s, a)$ for each newly encountered $(s, a)$ when using *proximal-policy extension* method. Therefore, by maintaining $\{M_\phi(s, a) | (s, a) \in \mathcal{D}^{cp}\}$ and updating it regularly, we can avoid recalculating $M_\phi(s, a)$ when using the *mixture distribution query*, thus saving a significant amount of overhead. The specific procedure is illustrated in Algorithm 1.

---

**Algorithm 1:** *Mixture Distribution Query*

**Input:** $\tau \in \mathcal{D}^{cp}$, $M_\phi(\tau)$, query size $b$ and mixture ratio $\kappa$.
**Output:** $\{\tau^0, \tau^1\}_{i=1}^b$

1 **for** $i = 1$ *to* $\kappa b$ **do**
2     $\tau^0, \tau^1 \sim P^{out}(\tau)$          // sample $\tau$ outside the distribution of $\mathcal{D}^p$
3 **for** $i = 1$ *to* $(1 - \kappa)b$ **do**
4     $\tau^0, \tau^1 \sim P^{in}(\tau)$           // sample $\tau$ inside the distribution of $\mathcal{D}^p$
5 **return** $\{\tau^0, \tau^1\}_{i=1}^b$

---

**Proximal Policy Exploration Algorithm** In summary, the *proximal-policy extension* method and the *mixture distribution query* method complement each other. The use of the *mixture distribution query* method can mitigate potential issues that might arise from solely using the *proximal-policy extension* method. The combination of these two methods forms our PPE algorithm, with the algorithmic process detailed in Algorithm 2.

In Algorithm 2, $\mathcal{D}^m = \{(s, a, M_\phi(s, a)) | (s, a) \in \mathcal{D}^{cp}\}$. The parts highlighted in brown represent the additions made by our algorithm compared to the basic algorithm framework.

Improvements in different algorithms typically focus on various stages: the data storage stage (Line 5, QPA (Hu et al., 2023)) , the data selection for querying stage (Line 7, QPA, B-Pref (Hu et al., 2023; Lee et al., 2021b)) , the reward model update stage (Line 12, SURF, PEBBLE (Park et al., 2022; Lee et al., 2021a)) , and the agent update stage (Line 17, RUNE, QPA (Liang et al., 2022; Hu et al., 2023)). Our approach, however, primarily enhances the data exploration stage, offering the advantage of excellent compatibility with existing methods. Although we use the *mixture distribution query* method for data selection, it does not conflict with existing query methods. We can apply the *mixture distribution query* method as a post-processing step on the results of existing query methods to select suitable data for querying. This further demonstrates the compatibility of our approach.

In practical applications, PPE can be implemented as an algorithmic plugin within an existing framework. This integration enhances the policy exploration process without requiring extensive modifications to the current framework.

---

**Algorithm 2:** Proximal Policy Exploration

---

**Input:** Query frequency $K$, feedback size once query $b$, mixture ratio $\kappa$ and morse buffer $\mathcal{D}^m$

1   Unsupervised pretraining            `// Lee et al. (2021b)`
2   **for** *each iteration* **do**
3     $a \sim \pi_E(\cdot|s)$     `// Sample action via` *proximal-policy extension*`, Eq.(4)`
4     $\{s, a, M_\phi(s,a)\} \cup \mathcal{D}^m$         `// Store the OOD metric of transition`
5     Store new transition $(s, a)$
6     **if** *iteration$\%K == 0$* **then**
7        $\left.\begin{array}{l} \{\tau^0, \tau^1\}_{i=1}^{(1-\kappa)b} \sim P^{in}(\tau) \\[2mm] \{\tau^0, \tau^1\}_{i=b+1}^{\kappa b} \sim P^{out}(\tau) \end{array}\right\}$   `//` *Mixture distribution query*`, Algorithm1`
8        Query for preference $\{y\}_{i=1}^b$
9        Store preference $\mathcal{D}^p \leftarrow \mathcal{D}^p \cup \{\tau^0, \tau^1, y\}_{i=1}^b$
10       **for** *each gradient step* **do**
11          Sample a minibatch preference $\mathcal{B} \leftarrow \{\tau^0, \tau^1, y\}_{i=1}^h \sim \mathcal{D}^p$
12          Training the reward model
13          Optimize loss of $M_\phi$ in Eq.(2) *w.r.t.* $\phi$ using $\mathcal{B}$
14        Relabel the reward in $\mathcal{D}$           `// Lee et al. (2021b)`
15        Relabel the OOD metric via $M_\phi(\cdot)$ for $(s, a) \in \mathcal{D}^{cp}$
16     **for** *each gradient step* **do**
17        Optimize $\pi_T$ via SAC method

---

# 4 EXPERIMENTS

Our method, as outlined in Section 3.2, is designed to be orthogonal and highly compatible with existing strategies. Notably, our *mixed distributed query* technique does not interfere with the *policy alignment query* employed in the QPA method. This compatibility allows us to seamlessly integrate PPE into the QPA algorithm for subsequent experiments. To simplify our discussion, we will directly refer to this integrated approach as PPE henceforth.

We conducted an evaluation of our method using the MetaWorld (Yu et al., 2020) and DMControl (Tassa et al., 2018) benchmarks. For a comprehensive comparison, we selected several baselines, including PEBBLE (Lee et al., 2021a), SURF (Park et al., 2022), RUNE (Liang et al., 2022), and the previous state-of-the-art method, QPA (Hu et al., 2023). In our experiments, we used five different seeds to compute the average performance. The shaded areas in the plots represent the 95% confidence intervals. For a complete understanding of our experimental details, please refer to Appendix I. Moreover, we also made use of the official code repositories provided in the papers of the corresponding baseline algorithms for a fair comparison.

## 4.1 BENCHMARK TASK PERFORMANCE

**Locomotion tasks in DMControl suite.** We selected six complex tasks from DMControl, namely *Walker-walk*, *Walker-run*, *Cheetah-run*, *Humanoid-stand*, *Quadruped-walk*, and *Quadruped-run*, to evaluate the performance of the PPE method. The dashed black line in our results represents the time step at which feedback collection was terminated.

Our method demonstrated superior performance across these tasks, as evidenced by the learning curve of PPE, which typically exhibited the steepest slope before the termination of feedback collection. This indicates that PPE can more effectively select and utilize feedback within a constrained quantity. Consequently, this validates our proposition that expanding the preference buffer coverage enhances the reward model's evaluation capabilities and makes policy updates more reliable.

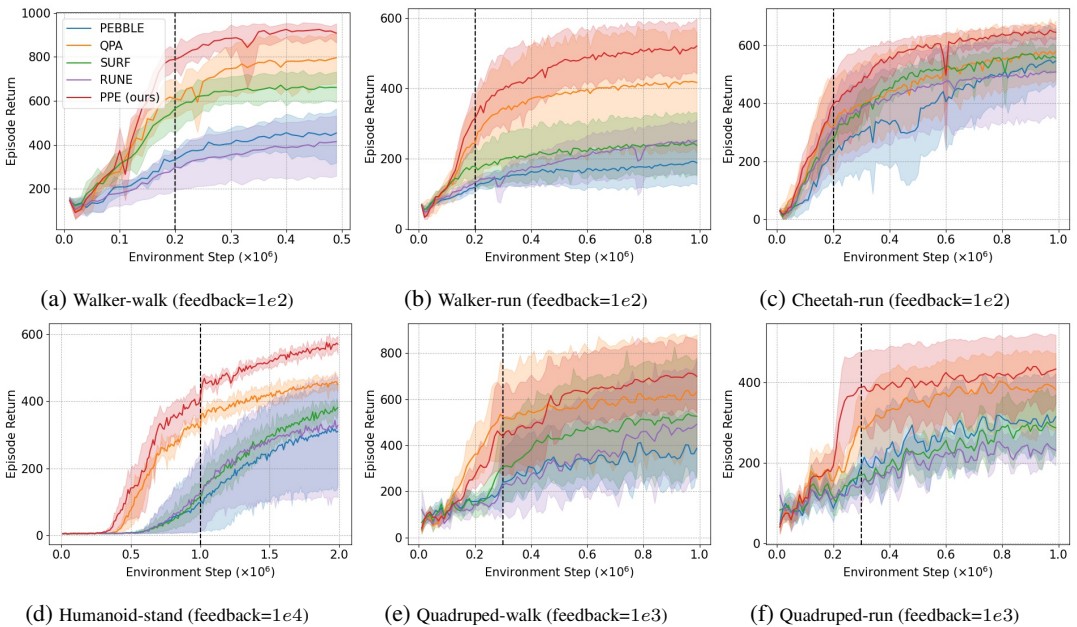

Figure 2: Learning curves for DMControl tasks, measured by the ground truth reward. The dashed black line marks the final feedback collection step.

**Robotic Manipulation Tasks in MetaWorld**    We conducted experiments on three complex manipulation tasks in MetaWorld: *Hammer*, *Sweep-into*, and *Drawer-open*. The learning curves for these tasks are presented in Figure 3. Similar to prior works (Christiano et al., 2017; Lee et al., 2021b; Park et al., 2022; Liang et al., 2022; Hu et al., 2023), we employed the ground truth success rate as a metric to quantify the performance of these methods.

Our results further demonstrate that PPE effectively enhances the feedback efficiency across a diverse range of complex tasks. However, we observed that while RUNE (Liang et al., 2022) did not perform well on DMControl tasks, it achieved performance second only to PPE on MetaWorld tasks. Additionally, we found that the performance variance of PbRL algorithms increases in the MetaWorld environment compared to DMControl. This phenomenon has also been observed in other PbRL literature (Lee et al., 2021b; Park et al., 2022; Hu et al., 2023; Liang et al., 2022). For a more detailed comparison, we provide additional numerical results in Appendix E.

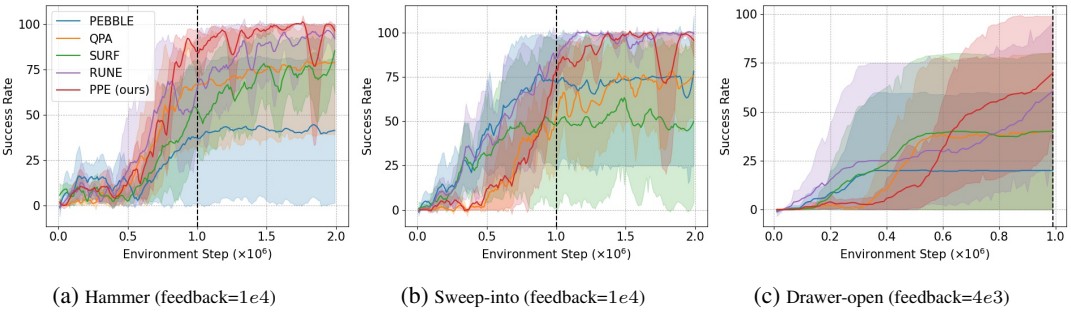

Figure 3: Learning curves for robotic manipulation tasks in MetaWorld, measured by the ground truth success rate. The dashed black line indicates the final feedback collection step.

## 4.2 ABLATION STUDY

To further investigate the impact of each component in PPE, we conducted additional ablation experiments on the Walker-walk task. These experiments aim to provide empirical evidence for the parameter selection of PPE.

To assess the roles of the *proximal-policy extension* method (EXT) and the *mixture distribution query* method (MDQ) within PPE, we incrementally applied these methods to the backbone algorithm QPA. As shown in Figure 4a, using either EXT or MDQ alone does not result in significant improvements. As described in Section 3.2, EXT and MDQ complement each other. Using only EXT increases the amount of out-of-preference buffer distribution data in the replay buffer without directly enhancing the coverage of the preference buffer. Conversely, using only MDQ fails to introduce sufficient out-of-preference buffer distribution data into the preference buffer due to the lack of active exploration, thus not effectively strengthening the reward model. Therefore, the superior performance of PPE arises from the mutual compensation of the shortcomings of EXT and MDQ.

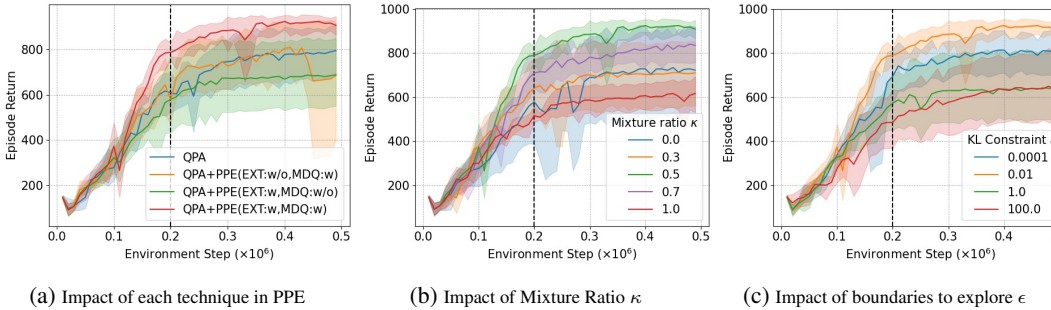

(a) Impact of each technique in PPE     (b) Impact of Mixture Ratio $\kappa$     (c) Impact of boundaries to explore $\epsilon$

Figure 4: Various ablation studies on the Walker-walk task, with the dashed black line indicating the final feedback collection step.

Next, we examine the effect of the mixture ratio $\kappa$ in MDQ using the complete PPE, which determines the balance of in-distribution and out-of-distribution data in the preference buffer. As Figure 4b shows, optimal performance is achieved at $\kappa = 0.5$. This result confirms that indiscriminate addition of out-of-distribution data to the preference buffer can overextend distribution boundaries, undermining the reward model's effectiveness that relies not just on preference buffer coverage, but also on data volume where policy evaluation is needed. Furthermore, exclusively sampling in-distribution data could hinder the reward model's adaptability to new distribution of trajectories following policy updates.

Additionally, we conducted ablation experiments on the KL constraint $\epsilon$, mentioned in Eq.(1). This parameter represents the exploration boundary of EXT for out-of-distribution data. As discussed in Appendix D, theoretically, using EXT requires that the behavior policy and target policy do not differ significantly. This implies that if $\epsilon$ is too large, performance cannot be theoretically guaranteed. Conversely, if $\epsilon$ is too small, EXT loses its exploratory significance. Experimental results, shown in Figure 4c, confirm this property: both excessively large and small values of $\epsilon$ negatively impact the results. Therefore, we recommend setting $\epsilon$ to 0.01.

## 5 RELATED WORK

**Human-in-the-loop Reinforcement Learning**    Human-in-the-loop reinforcement learning (RL) uses human preferences to train RL agents, allowing humans to specify desired behaviors through comparative judgments (Akrour et al., 2011; Pilarski et al., 2011; Christiano et al., 2017; Stiennon et al., 2020; Wu et al., 2021). However, acquiring these preferences is costly and requires high feedback efficiency (Lee et al., 2021a; Park et al., 2022; Liang et al., 2022; Liu et al., 2024b).

**Query Selection Schemes in PbRL**    Query selection schemes are crucial in preference-based RL (PbRL) for improving feedback efficiency. Previous research has used metrics like entropy (Bıyık & Sadigh, 2018; Ibarz et al., 2018; Lee et al., 2021a), L2 distance in feature space (Bıyık et al.,

2020), and ensemble disagreement of the reward model (Christiano et al., 2017; Ibarz et al., 2018; Lee et al., 2021a; Park et al., 2022; Liang et al., 2022) to evaluate query quality. These metrics guide sampling strategies such as greedy sampling (Biyik & Sadigh, 2018), the K-medoids algorithm (Biyik & Sadigh, 2018; Rdusseeun & Kaufman, 1987), and Poisson disk sampling (Bridson, 2007; Bıyık et al., 2020) to identify the most "informative" queries.

However, Hu et al. (2023) argued that these methods offer limited benefits to policy learning. They identified the issue of *Query-policy Misalignment* in PbRL and proposed the *query-policy align* method to address it. Our *mixture distribution query* method complements existing approaches and can be seamlessly integrated with them. In the experiments, we combined our method with Hu et al. (2023)'s method to further improve the query selection scheme.

**Exploration in Reinforcement Learning**   The trade-off between exploitation and exploration is critical in reinforcement learning (RL) (Sutton, 2018; Hao et al., 2024). Exploration algorithms are designed to encourage RL agents to visit a wide range of states. Notable methods include uncertainty-driven exploration approaches (Bellemare et al., 2016; Tang et al., 2017; Ciosek et al., 2019; Bai et al., 2021; Liu et al., 2024a), intrinsic-reward driven approaches (Ostrovski et al., 2017; Houthooft et al., 2016; Pathak et al., 2017; Bai et al., 2023), and others (Hazan et al., 2019; Liu & Abbeel, 2021). In PbRL, Liang et al. (2022) introduced an intrinsic reward to drive exploration by leveraging reward model disagreements, aligning exploration with human preferences. Our work also focuses on PbRL exploration, aiming to collect diverse data for the preference buffer to build a more reliable reward model. Unlike Liang et al. (2022), we emphasize the importance of preference buffer coverage for constructing a reward model.

## 6   CONCLUSION AND DISCUSSION

This paper highlights the critical role of preference buffer coverage in the evaluative accuracy of reward models. Our findings indicate that a reward model's accuracy is the highest for trajectories within the preference buffer's distribution and significantly decreases for out-of-distribution trajectories. We introduce PPE algorithm, which actively expands the preference buffer coverage to enhance the reliability of the reward model, comprising two complementary components: the *proximal-policy extension* method and the *mixture distribution query* method. These components synergistically work to expand the preference buffer coverage while balancing the inclusion of both in-distribution and out-of-distribution data. PPE provides a more reliable reward model, thereby reducing the potential of misleading policy improvements. PPE has demonstrated substantial gains in feedback and sample efficiency through extensive evaluations on the DMControl and MetaWorld benchmarks. These results underscore the importance of actively expanding preference buffer coverage in PbRL research.

In this study, our main focus is on enhancing the reward model's quality by actively expanding the preference buffer's coverage. However, our current query method does not consider the variations in information between different pairs of agent behaviors. As we advance our research, we plan to investigate advanced methods to boost feedback efficiency. We believe that considering factors such as data similarity and clustering traits can further refine and optimize our query method.

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

## A    THE PROCESS OF REWARD MODEL TRAINING IN PbRL

Using a preference dataset $\mathcal{D}_p$, the reward model $\hat{r}_\psi$ learns to assign higher proxy returns $\hat{G}_\psi = \sum_t \hat{r}_\psi(s_t, a_t)$ to preferred trajectories. Employing the Bradley-Terry model (Bradley & Terry, 1952), the probability that one trajectory is preferred over another is computed as:

$$P_\psi(\tau^1 \succ \tau^0) = \frac{\exp\left(\sum_t \hat{r}_\psi(s_t^1, a_t^1)\right)}{\sum_{i \in \{0,1\}} \exp\left(\sum_t \hat{r}_\psi(s_t^i, a_t^i)\right)}. \tag{7}$$

The probability estimate $P_\psi$ is used to minimize the cross-entropy between the predicted and true preference labels:

$$L_{CE} = -\mathbb{E}_{(\tau^0, \tau^1, y_p) \sim \mathcal{D}_p} \left[ \mathbb{I}\{y_p = (\tau^0 \succ \tau^1)\} \log P_\psi(\tau^0 \succ \tau^1) + \mathbb{I}\{y_p = (\tau^1 \succ \tau^0)\} \log P_\psi(\tau^1 \succ \tau^0) \right]. \tag{8}$$

After optimizing the reward function $\hat{r}_\psi$ from human preferences, PbRL algorithms enable training of RL agents with standard RL algorithms, treating the proxy rewards from $\hat{r}_\psi$ as if they were ground truth rewards from the environment.

## B    INFORMATION ABOUT MORSE NEURAL NETWORK

**Definition 1 (Morse Kernel)** *A Morse Kernel is a positive definite kernel $K$. When applied in a space $Z = \mathbb{R}^k$, the kernel $K(z_1, z_2)$ takes values in the interval $[0, 1]$ and satisfies $K(z_1, z_2) = 1$ if and only if $z_1 = z_2$.*

All kernels of the form $K(z_1, z_2) = e^{-D(z_1, z_2)}$, where $D(\cdot, \cdot)$ is a divergence (Amari, 2016), are considered Morse Kernels. In this study, we utilize the Radial Basis Function (RBF) Kernel,

$$K_{RBF}(z_1, z_2) = e^{-\frac{\lambda^2}{2} \|z_1 - z_2\|^2}, \tag{9}$$

where $\lambda$ is a scale parameter of the kernel (Seeger, 2004).

Consider a neural network that maps from a feature space $X$ to a latent space $Z$ via a function $f_\phi : X \to Z$, with parameters $\phi$. Here, $X \in \mathbb{R}^d$ and $Z \in \mathbb{R}^k$. A Morse Kernel can be used to impose structure on the latent space.

**Definition 2 (Morse Neural Network)** *A Morse neural network is defined as a function $f_\phi : X \to Z$ combined with a Morse Kernel $K(z, t)$, where $z \subset Z$ is a target chosen as a hyperparameter of the model. The Morse neural network is expressed as $M_\phi(x) = 1 - K(f_\phi(x), t)$.*

According to Definition 1, $M_\phi(x)$ takes values in the interval $[0, 1]$. When $M_\phi(x) = 0$, $x$ corresponds to a mode that aligns with the level set of the submanifold of the Morse neural network. Additionally, $1 - M_\phi(x)$ represents the certainty that the sample $x$ is from the training dataset, making $M_\phi(x)$ a measure of the epistemic uncertainty of $x$.

The function $-\log[1 - M_\phi(x)]$ quantifies a squared distance, $d(\cdot, \cdot)$, between $f_\phi(x)$ and the nearest mode in the latent space at $m$:

$$d(z) = \min_{m \in M} d(z, m), \tag{10}$$

where $M$ is the set of all modes. This provides information about the topology of the submanifold and satisfies the Morse–Bott non-degeneracy condition (Basu & Prasad, 2023).

The Morse neural network exhibits the following properties:

1. $M_\phi(x) \in [0, 1]$;
2. $M_\phi(x) = 0$ at its mode submanifolds;
3. $-\log[1 - M_\phi(x)] \geq 0$ represents a squared distance that satisfies the Morse–Bott non-degeneracy condition on the mode submanifolds;
4. Since $M_\phi(x)$ is an exponentiated squared distance, the function is distance-aware, meaning that as $f_\phi(x) \to t$, $[1 - M_\phi(x)] \to 1$.

## C  DERIVATION OF THE LOSS FUNCTION FOR MORSE NEURAL NETWORK IN PBRL

We achieve the measurement of whether the current data is outside the distribution of $\mathcal{D}^p$ using the Morse Neural Network by minimizing the KL divergence $D_{KL}(\mathcal{D}^p(s,a)\|1-M_\phi(s,a))$. The detailed derivation process is as follows:

$$\min_\phi \mathop{\mathbb{E}}_{s,a\sim\mathcal{D}^p}\left[\log\frac{\mathcal{D}^p(s,a)}{1-M_\phi(s,a)}\right] + \mathop{\mathbb{E}}_{s\sim\mathcal{D}^p}\left[\frac{1}{|\mathcal{A}|}\int_{a\in\mathcal{A}}1-M_\phi(s,a)-\mathcal{D}^p(s,a)da\right].$$

$$\rightarrow\min_\phi \mathop{\mathbb{E}}_{s,a\sim\mathcal{D}^p}\left[-\log\left[1-M_\phi(s,a)\right]+\mathop{\mathbb{E}}_{a_u\sim\text{Uniform}(\mathcal{A})}\left[1-M_\phi(s,a)\right]\right].$$

$$\rightarrow\min_\phi\frac{1}{N}\sum_{s,a\sim\mathcal{D}^p}\left[-\log K_{RBF}(f_\phi(s,a),a)+\frac{1}{M}\sum_{a_u\sim\text{Uniform}(\mathcal{A})}K_{RBF}(f_\phi(s,a_u),a_u)\right]. \tag{11}$$

$$\rightarrow\min_\phi\frac{1}{N}\sum_{s,a\sim\mathcal{D}^p}\left[\frac{\lambda^2}{2}\|f_\phi(s,a)-a\|^2+\frac{1}{M}\sum_{a_u\sim\text{Uniform}(\mathcal{A})}\exp^{-\frac{\lambda^2}{2}\|f_\phi(s,a_u)-a_u\|^2}\right].$$

## D  PROOF OF PROPOSTION 1

Consider the formula for the KL divergence between two high-dimensional Gaussian distributions:

$$D_{KL}(\mathcal{N}(\mu,\Sigma),\mathcal{N}(\mu_T,\Sigma_T))=\frac{1}{2}\left[(\mu-\mu_T)^\mathrm{T}\Sigma_T^{-1}(\mu-\mu_T)-\log\det(\Sigma_T^{-1}\Sigma)+tr(\Sigma_T^{-1}\Sigma)-n\right]. \tag{12}$$

When $D_{KL}(\mathcal{N}(\mu,\Sigma),\mathcal{N}(\mu_T,\Sigma_T))\leq\epsilon$ is employed as a constraint, the solution to the optimization problem $\arg\max_{\mu,\Sigma}\mathbb{E}_{a\sim\mathcal{N}(\mu,\Sigma)}[M_\phi(s,a)]$ is typically achieved through iterative means. However, considering our objective for the calculated $\mu,\Sigma$ to more effectively explore data from the out-of-preference buffer distribution within the proximal policy region, and the real-time requirement for problem-solving with each agent-environment interaction, we propose a more efficient closed-form approximation to the original problem by appropriately tightening the constraint, as shown in Proposition 1.

We introducing $\Sigma=\Sigma_T$, and the tightened constraint can be expressed as:

$$D_{KL}(\mathcal{N}(\mu,\Sigma_T),\mathcal{N}(\mu_T,\Sigma_T))\leq\epsilon.$$

$$\rightarrow\frac{1}{2}\left[(\mu-\mu_T)^\mathrm{T}\Sigma_T^{-1}(\mu-\mu_T)-\log\det(\Sigma_T^{-1}\Sigma_T)+tr(\Sigma_T^{-1}\Sigma_T)-n\right]\leq\epsilon. \tag{13}$$

$$\rightarrow\frac{1}{2}\left[(\mu-\mu_T)^\mathrm{T}\Sigma_T^{-1}(\mu-\mu_T)\right]\leq\epsilon.$$

Substituting this into Eq.(3), we derive a simplified optimization problem:

$$\max_\mu\mathop{\mathbb{E}}_{a\sim\mathcal{N}(\mu,\Sigma_T)}[M_\phi(s,a)],$$

$$\text{s.t.}(\mu-\mu_T)^\mathrm{T}\Sigma_T^{-1}(\mu-\mu_T)\leq 2\epsilon. \tag{14}$$

To address the problem in Eq.(14), we construct the following Lagrangian function:

$$L=M_\phi(s,a)-\xi((\mu-\mu_T)^\mathrm{T}\Sigma_T^{-1}(\mu-\mu_T)-2\epsilon). \tag{15}$$

Deriving with respect to $\mu$ yields:

$$\nabla_\mu L=\nabla_a M_\phi(s,a)|_{a=\mu}-\xi\Sigma_T^{-1}(\mu-\mu_T). \tag{16}$$

Setting $\nabla_\mu L=0$, we find:

$$\mu=\mu_T+\frac{1}{\xi}\Sigma_T\,\nabla_a M_\phi(s,a)|_{a=\mu}. \tag{17}$$

By applying the KKT conditions, we deduce:

$$(\mu - \mu_T)^{\mathrm{T}} \Sigma_T^{-1} (\mu - \mu_T) - 2\epsilon = 0. \tag{18}$$
$$\xi > 0.$$

Further, via plugging Eq.(17) in Eq.(18), we can solve to obtain:

$$\frac{1}{\xi^2} \left( \Sigma_T \left. \nabla_a M_\phi(s,a) \right|_{a=\mu} \right)^{\mathrm{T}} \Sigma_T^{-1} \left( \Sigma_T \left. \nabla_a M_\phi(s,a) \right|_{a=\mu} \right) = 2\epsilon, \ \xi > 0.$$

$$\rightarrow \xi^2 = \frac{[\nabla_a M_\phi(s,a)]_{a=\mu}^{\mathrm{T}} \, \Sigma_T \, [\nabla_a M_\phi(s,a)]_{a=\mu}}{2\epsilon}, \ \xi > 0. \tag{19}$$

$$\rightarrow \xi = \sqrt{\frac{[\nabla_a M_\phi(s,a)]_{a=\mu}^{\mathrm{T}} \, \Sigma_T \, [\nabla_a M_\phi(s,a)]_{a=\mu}}{2\epsilon}}.$$

Through Eq.(19), we find that $\xi$ is a function of $\mu$. However, Eq.(17) is a differential equation, which is challenging to solve directly for $\mu$. Therefore, we perform a Taylor expansion on $[\nabla_a M_\phi(s,a)]_{a=\mu}$:

$$\left. \nabla_a M_\phi(s,a) \right|_{a=\mu} \approx \left. \nabla_a M_\phi(s,a) \right|_{a=\mu_T} + \left. \nabla_a^2 M_\phi(s,a) \right|_{a=\mu_T} (\mu - \mu_T). \tag{20}$$

This implies that when $\mu$ is sufficiently close to $\mu_T$, we can approximate:

$$\left. \nabla_a M_\phi(s,a) \right|_{a=\mu} \approx \left. \nabla_a M_\phi(s,a) \right|_{a=\mu_T}. \tag{21}$$

Since our goal is to increase the density of proximal policy data in the preference buffer, thereby enhancing the reward model's evaluation capability under the current policy distribution, this approximation does not conflict with our objective and is indeed very fitting.

Thus, further, we can deduce:

$$\mu \approx \mu_T + \frac{\sqrt{2\epsilon} \cdot \Sigma_T [\nabla_a M_\phi(s,a)]_{a=\mu_T}}{\sqrt{[\nabla_a M_\phi(s,a)]_{a=\mu_T}^{\mathrm{T}} \Sigma_T [\nabla_a M_\phi(s,a)]_{a=\mu_T}}}. \tag{22}$$

Therefore, the exploration behavior policy $\mathcal{N}(\mu_E, \Sigma_E)$ can be expressed as

$$\mu_E = \mu_T + \frac{\sqrt{2\epsilon} \cdot \Sigma_T [\nabla_a M_\phi(s,a)]_{a=\mu_T}}{\sqrt{[\nabla_a M_\phi(s,a)]_{a=\mu_T}^{\mathrm{T}} \Sigma_T [\nabla_a M_\phi(s,a)]_{a=\mu_T}}}, \text{and } \Sigma_E = \Sigma_T. \tag{23}$$

# E  ADDITIONAL EXPERIMENTS

| Task | PEBBLE | SURF | RUNE | QPA | PPE |
|------|--------|------|------|-----|-----|
| Walker-walk_1e2 | $453.43 \pm 159.43$ | $661.01 \pm 91.72$ | $414.62 \pm 182.16$ | $796.08 \pm 147.94$ | $\mathbf{908.09 \pm 55.30}$ |
| Walker-run_1e2 | $188.21 \pm 79.86$ | $237.65 \pm 116.85$ | $251.48 \pm 104.98$ | $416.52 \pm 222.01$ | $\mathbf{520.18 \pm 101.72}$ |
| Quadruped-walk_1e3 | $369.51 \pm 134.22$ | $488.71 \pm 283.49$ | $440.30 \pm 296.02$ | $567.80 \pm 291.57$ | $\mathbf{660.07 \pm 175.58}$ |
| Quadruped-run_1e3 | $314.91 \pm 120.87$ | $287.37 \pm 101.75$ | $231.85 \pm 60.14$ | $382.03 \pm 123.60$ | $\mathbf{433.42 \pm 116.58}$ |
| Cheetah-run_1e2 | $545.77 \pm 130.00$ | $556.78 \pm 59.323$ | $508.60 \pm 186.06$ | $578.89 \pm 133.14$ | $\mathbf{644.91 \pm 30.37}$ |
| Humanoid-stand_1e4 | $306.08 \pm 171.92$ | $377.51 \pm 20.35$ | $351.10 \pm 197.75$ | $455.81 \pm 25.99$ | $\mathbf{577.12 \pm 30.93}$ |
| Drawer-open_4e3 | $20.00 \pm 44.72$ | $40.09 \pm 54.89$ | $48.45 \pm 47.95$ | $40.09 \pm 54.89$ | $\mathbf{69.81 \pm 43.41}$ |
| Sweep-into_1e4 | $62.58 \pm 57.44$ | $40.06 \pm 50.80$ | $\mathbf{99.62 \pm 0.56}$ | $80.67 \pm 27.00$ | $96.47 \pm 8.47$ |
| Hammer_1e4 | $41.31 \pm 53.57$ | $85.23 \pm 26.18$ | $91.86 \pm 17.77$ | $78.75 \pm 44.04$ | $\mathbf{96.27 \pm 5.19}$ |

Table 1: Performance of benchmark experiments

**Performance of Benchmark Tasks**  We recorded the performance of different algorithms—QPA, PEBBLE, SURF, RUNE, and PPE—on DMControl and MetaWorld in Table 1. Each value represents the mean and variance calculated from the last five evaluations under different seeds for the same algorithm.

**Exploration Methods Across Different Backbones**    As shown in Tables 2 and 3, we applied PPE and RUNE to QPA and PEBBLE, respectively. This approach not only verifies the compatibility of PPE but also highlights the performance differences of various exploration methods across different backbones.

| Task | PEBBLE | PEBBLE+RUNE | PEBBLE+PPE |
|------|--------|-------------|------------|
| Walker-walk_1e2 | $453.43 \pm 159.43$ | $414.62 \pm 182.16$ | $\mathbf{499.73 \pm 82.75}$ |
| Walker-run_1e2 | $188.21 \pm 79.86$ | $251.48 \pm 104.98$ | $\mathbf{257.64 \pm 58.59}$ |
| Quadruped-walk_1e3 | $369.51 \pm 134.22$ | $440.30 \pm 296.02$ | $\mathbf{451.06 \pm 223.27}$ |
| Quadruped-run_1e3 | $314.91 \pm 120.87$ | $231.85 \pm 60.14$ | $\mathbf{373.09 \pm 149.10}$ |
| Cheetah-run_1e2 | $545.77 \pm 130.00$ | $508.60 \pm 186.06$ | $\mathbf{569.54 \pm 84.27}$ |
| Humanoid-stand_1e4 | $306.08 \pm 171.92$ | $351.10 \pm 197.75$ | $\mathbf{357.13 \pm 76.15}$ |

Table 2: The Performance of Different Exploration Methods on PEBBLE

| Task | QPA | QPA+RUNE | QPA+PPE |
|------|-----|----------|---------|
| Walker-walk_1e2 | $796.08 \pm 147.94$ | $704.39 \pm 133.45$ | $\mathbf{908.09 \pm 55.30}$ |
| Walker-run_1e2 | $416.52 \pm 222.01$ | $429.66 \pm 173.62$ | $\mathbf{520.18 \pm 101.72}$ |
| Quadruped-walk_1e3 | $567.80 \pm 291.57$ | $593.61 \pm 295.84$ | $\mathbf{660.07 \pm 175.58}$ |
| Quadruped-run_1e3 | $382.03 \pm 123.60$ | $367.71 \pm 108.01$ | $\mathbf{433.42 \pm 116.58}$ |
| Cheetah-run_1e2 | $578.89 \pm 133.14$ | $\mathbf{689.52 \pm 49.39}$ | $644.91 \pm 30.37$ |
| Humanoid-stand_1e4 | $455.81 \pm 25.99$ | $419.74 \pm 27.38$ | $\mathbf{577.12 \pm 30.93}$ |

Table 3: The Performance of Different Exploration Methods on QPA

**Ablation Study on $\kappa$**    Under the Walker-walk experiment setting with 100 feedback instances, we investigated the impact of the mixture ratio $\kappa$ on the experimental results, as shown in Table 4. Based on these results, we set the mixture ratio $\kappa$ to 0.5 for all subsequent experiments.

| $\kappa$ | Episode Return | $\kappa$ | Episode Return | $\kappa$ | Episode Return |
|------|---------|------|---------|------|---------|
| 0.0 | $722.33 \pm 256.97$ | 0.4 | $756.41 \pm 215.27$ | 0.8 | $696.62 \pm 243.53$ |
| 0.1 | $795.26 \pm 174.23$ | 0.5 | $\mathbf{908.09 \pm 55.30}$ | 0.9 | $744.50 \pm 173.46$ |
| 0.2 | $688.53 \pm 212.85$ | 0.6 | $714.03 \pm 230.37$ | 1.0 | $616.22 \pm 106.39$ |
| 0.3 | $710.22 \pm 187.74$ | 0.7 | $834.91 \pm 103.28$ | | |

Table 4: Impact of Mixture Ratio $\kappa$ on Walker-walk performance with 100 feedback instances

**Ablation Study on the Various Components of PPE**    We denote the *proximal-policy extension* method as EXT and the *mixture distribution query* method as MDQ. The specific details are recorded in Table 5.

**Ablation Study on KL Constraint $\epsilon$**    In Table 6, we present the impact of different KL constraints $\epsilon$ on the performance

# F    ABOUT OOD DETECTION COMPUTATIONAL COST

## F.1    DISCUSSION ON $f_\phi$

Firstly, In our study, we utilized a neural network with a 3x256 architecture to learn the function $f_\phi$ required for the Morse network, as described in Eq.(4).

Secondly, we do not rely on the specific outputs of the Morse network to determine whether data is OOD. Instead, we only utilize the gradient $\nabla_a M_\phi(s, a)$ and use it as a basis for sampling data

| Algo | Episode Return | Algo | Episode Return |
|------|----------------|------|----------------|
| QPA | $796.08 \pm 147.94$ | QPA+PPE(EXP:w,MDQ:w) | $\mathbf{908.09 \pm 55.30}$ |
| QPA+PPE(EXP:w,MDQ:w/o) | $689.33 \pm 194.50$ | QPA+PPE(EXP:w/o,MDQ:w) | $685.03 \pm 346.27$ |

Table 5: Impact of various components of PPE on Walker-walk performance with 100 feedback instances

| KL Constraint $\epsilon$ | Episode Return | KL Constraint $\epsilon$ | Episode Return |
|--------------------------|----------------|--------------------------|----------------|
| 1e-4 | $806.67 \pm 137.13$ | 1e-2 | $\mathbf{908.09 \pm 55.30}$ |
| 1e-1 | $745.54 \pm 163.10$ | 1e0 | $638.53 \pm 202.73$ |
| 1e1 | $783.26 \pm 163.70$ | 1e2 | $635.46 \pm 214.01$ |

Table 6: Impact of various KL Constraint $\epsilon$ on Walker-walk performance with 100 feedback instances

in the '*Mixture Distribution Query*'. These applications do not demand high precision in the Morse network's outputs; they only require a relative distinction in magnitude between in-distribution and out-of-distribution data.

Lastly, Given that our dataset is not very large, especially when QPA is used as the backbone with a dataset size of only '$10 \times$ episode_length', which does not impose significant stress on the neural network.

Considering computational costs, we only train the Morse network for an additional 200 iterations after completing after per query. It is noteworthy that in many tasks, QPA and SURF involve training the reward model thousands of times after per query. Therefore, our use of the Morse network effectively meets our needs without incurring substantial additional computational overhead.

### F.2 EXPERIMENTS RESULTS OF COMPUTATIONAL COST

We averaged the time required for QPA and QPA+PPE to train the reward model (and the Morse network) after five query phases on the walker_walk task, all conducted on the same machine.

| Method | Average Time (seconds) |
|--------|------------------------|
| QPA | 45.29 |
| QPA+PPE | 50.42 |

Table 7: Average time comparison between QPA and QPA+PPE.

While PPE does introduce additional computational overhead, training the Morse network, like the reward model, is only necessary after each query. The total number of queries varies by task. For instance, in the 'walker_walk' task, we followed the QPA setup, requiring a total of 100 preference feedbacks, with each query obtaining 10 preference feedbacks. Therefore, the overall training process does not significantly increase computational cost.

## G MORE DETAILS ABOUT THE MOTIVATING EXAMPLE IN SECTION 3.1

### G.1 MEANING OF REGION $1 - 9$

In Section 3.1, "region 1-9" refers to square regions depicted in Figure 1a, with the lower-left corner as the origin. The grid is labeled from 0 to 9 on both the horizontal and vertical axes, increasing

from left to right and from bottom to top, respectively. For example, "region 3" denotes a grid area bounded by the segments from 0 to 3 on both axes.

## G.2 THE EVALUATION REGION FOR USED IN FIGURE 1D

The evaluation region used is the same as the training region. This figure is intended to explore how varying the amount of preference feedback affects the performance of the reward model when both the evaluation and training regions are fixed.

# H COVERAGE VISUALIZATION

We collected 100 feedback instances during the learning process of the Walker_walk task using the QPA and QPA+PPE methods. The state and action spaces of these (s, a) pairs were clustered into 10 and 20 groups, respectively, using KMeans. We then used heatmaps to illustrate how the coverage of the preference buffer changes as feedback increases.

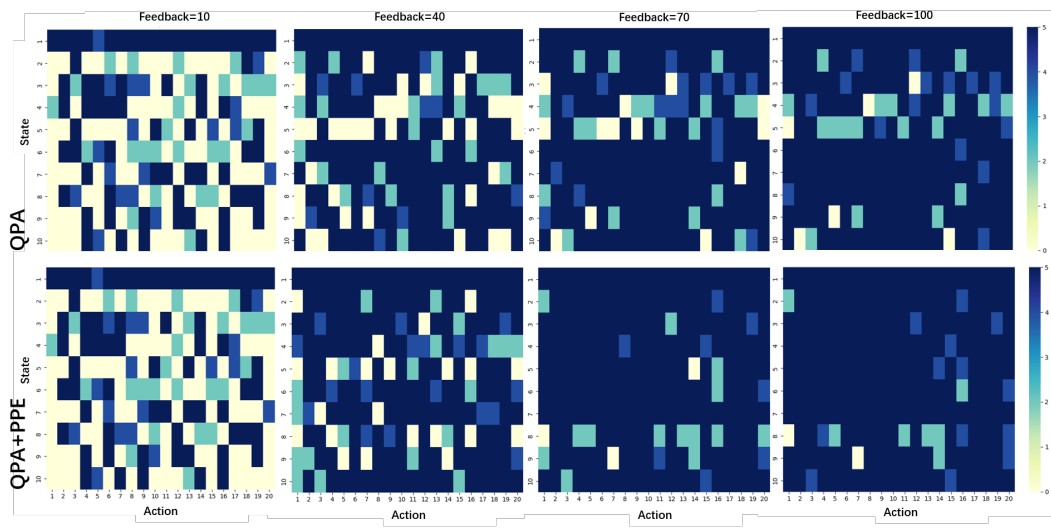

Figure 5: Distribution of actions in different discrete states after clustering. The horizontal axis represents the 20 clustered actions, and the vertical axis represents the 10 clustered states. The first and second rows show the changes in coverage of the preference buffer during training for the QPA and QPA+PPE methods, respectively.

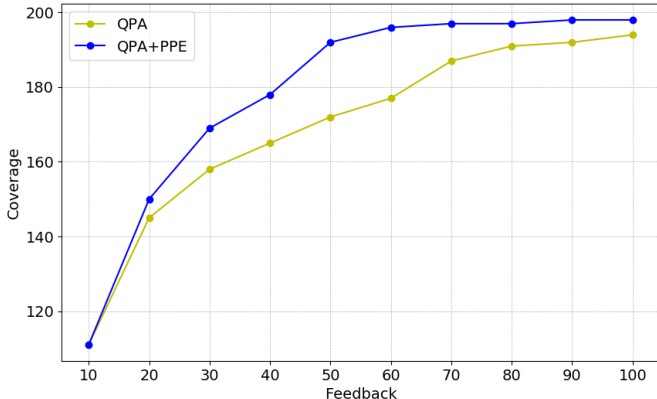

Figure 6: Changes in coverage of the 10×20 clustered (s, a) space for the preference buffer corresponding to the QPA and QPA+PPE methods under the same seed.

Figures 5 and 6 demonstrate that as the number of queries increases, the use of PQPA+PPE clearly enhances coverage compared to QPA.

# I  IMPLEMENTATION DETAILS

## I.1  FUNDAMENTAL PROCESS OF PBRL

An overview of the components in a typical PbRL setup can be provided as below:

    a).  Data collection

    b).  Data selection and preference labeling

    c).  Learning the reward model using preference labels $(\tau^0, \tau^1, y_p)$

    d).  Optimizing $\pi_T$ with the learned reward model via reinforcement learning methods

## I.2  ABOUT BUFFERS

### I.2.1  FUNCTIONS OF VARIOUS BUFFERS

- $\mathcal{D}^{cp}$ stores potential segments $\tau$ that might be selected during the "data selection and preference labeling" phase. Specifically, when selecting $(\tau^0, \tau^1)$ for preference labeling, these segments are drawn from $\mathcal{D}^{cp}$.

- $\mathcal{D}$ is the replay buffer, a fundamental concept in reinforcement learning, storing $(s_t, a_t, \hat{r}_t, s_{t+1})$ instead of the ground truth $r_t$ . It is used during the policy optimization phase with the learned reward model.

- $\mathcal{D}^p$ stores preference feedbacks $(\tau^0, \tau^1, y_p)$ for learninig the reward model.

- $\mathcal{D}^m$ stores an additional one-dimensional data $M_\phi(s, a)$ for each $(s, a)$ in $\mathcal{D}^{cp}$, as shown in Eq. 5. It is used to compute to assess the OOD degree of $\tau$.

### I.2.2  MEMORY USAGE

- $\mathcal{D}$ is essential for all off-policy reinforcement learning algorithms as a replay buffer.

- $\mathcal{D}^{cp}$ and $\mathcal{D}^p$ are necessary for existing online PbRL methods.

- $\mathcal{D}^m$ only requires storing an additional one-dimensional value $M_\phi(s, a)$ for each $(s, a)$ in $\mathcal{D}^{cp}$, which is a minor addition performed in Algorithm 2, line 4

Therefore, PPE does not require significantly more memory compared to previous online PbRL methods.

## I.3  ORIGIN OF THE CODE FOR BASELINE ALGORITHMS

To ensure fairness in our experiments, we used the original source code provided by the authors of each baseline algorithm. Specifically, the sources are as follows:

- PEBBLE, SURF:
  https://openreview.net/attachment?id=TfhfZLQ2EJO&name=supplementary_material

- RUNE:https://github.com/rll-research/rune

- QPA:https://github.com/huxiao09/QPA

- B-Pref: https://github.com/rll-research/BPref

The only modification we made was to unify the logging format during training. We changed QPA's logging from using wandb to the storage format used by the B-Pref framework, which is also used by PEBBLE, SURF, and RUNE.

### I.4 HUMAN INVOLVEMENT

In stage **b**, algorithms typically select $(\tau^0, \tau^1)$ pairs, which are then submitted for human preference labeling. In most PbRL implementations, scripts are typically used to simulate human preference labeling. Our paper follows the same setup.

The Mixture Distribution Query is used only in stage b to select , as shown in Algorithm 1. These selected pairs are then submitted for human preference labeling (Algorithm 2, line 8). This is the only stage that requires human involvement.

This process is consistent with what is described in PEBBLE (Algorithm 2, line 11), QPA [5] (Algorithm 1, line 6), and RUNE (Algorithm 1, line 9).

### I.5 HOW WERE PREFERENCES ELICITED?

We used the same approach as PEBBLE, SURF, RUNE, and QPA, utilizing the B-pref framework Shin et al. (2023) to script access to the ground truth reward, thereby simulating human preference labels.

### I.6 HOW TO OBTAIN GENUINE HUMAN PREFERENCES ONLINE

**Collecting Human Feedback**

```python
import imageio as iio

def get_label(self, sa_t_1, sa_t_2, physics_seg1, physics_seg2):

    frame_height, frame_width, channels = physics_seg1[0,0].shape

    # Create a video writer
    output_width = frame_width * 2  # The merged width is twice the original.
    output_height = frame_height
    fps = 30  # Set the frame rate.

    # Save video
    human_labels = np.zeros(sa_t_1.shape[0])
    for seg_index in range(physics_seg1.shape[0]):
        # render the pairs of segments and save the video
        # Create a video writer using imageio
        with iio.get_writer(f'output.mp4', fps=fps) as writer:
            # Iterate over all frames.
            for frame0, frame1 in zip(physics_seg1[seg_index], physics_seg2[seg_index]):
                # Horizontally merge frames
                combined_frame = np.hstack((frame0, frame1))
                # Write to the video file
                writer.append_data(combined_frame)
        labeling = True
        # provide labeling instruction and query human for preferences
        while(labeling):
            print("\n")
            print("--------------------------------------------------")
            print("Feedback number:", seg_index)
            # preference:
            # 0: segment 0 is better
            # 1: segment 1 is better
            while True:
                # check if it is 0/1/number type preference
                try:
                    rational_label = input("Preference: 0 or 1 or other number")
                    rational_label = int(rational_label)
                    break
                except:
                    print("Wrong label type. Please enter 0/1/other number.")
            print("--------------------------------------------------")
            human_labels[seg_index] = rational_label
            labeling = False
    #remove the hard-to-judge pairs of segments
    cancel = np.where((human_labels != 0) & (human_labels != 1))[0]
    human_labels = np.delete(human_labels, cancel, axis=0)
    sa_t_1 = np.delete(sa_t_1, cancel, axis=0)
    sa_t_2 = np.delete(sa_t_2, cancel, axis=0)
    print("valid query number:", len(human_labels))
    return sa_t_1, sa_t_2, human_labels.reshape(-1,1)
```

We achieve authentic interaction with humans in the process of obtaining human preferences through the code above. This involves presenting two sets of behavior segment videos to humans and requesting preference labels from them. The specific interaction interface is shown in Figure 7.

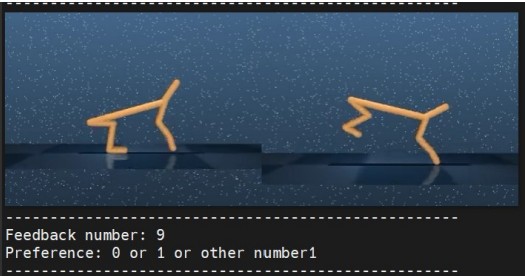

Figure 7: Through this interface, humans can provide preference labels for the agent's behavior.

### I.7 PARAMETER FOR ALGORITHMS

Our method does not introduce many additional parameters, as shown in Table 8. In this work, $\epsilon$ represents the KL divergence constraint between the behavior policy and the target policy in Eq.(3), which determines the exploration boundary in our approach. The parameter $\lambda$ controls the sensitivity of the Morse Neural Network. Lastly, $\kappa$, mentioned in Algorithm 2, is the mixture ratio that controls the proportion of samples drawn from each distribution.

| Hyper-parameter | Value | Hyper-parameter | Value |
|---|---|---|---|
| KL constraint $\epsilon$ | 1e-2 | Parameter for OOD detection $\lambda$ | 5 |
| Mixture ratio $\kappa$ | 0.5 | | |

Table 8: The hyperparameters of PPE

Additionally, we followed the parameter settings from the baseline papers (Hu et al., 2023; Lee et al., 2021b; Park et al., 2022; Liang et al., 2022; Lee et al., 2021a). The specific parameter configurations are detailed in Tables 9, 10, 11, and 12.

| Hyper-parameter | Value | Hyper-parameter | Value |
|---|---|---|---|
| Discount | 0.99 | Init temperature | 0.1 |
| Alpha learning rate | 1e-4 | Batch size | 1024 |
| Critic target update freq | 2 | Critic EMA | 5e-3 |
| Critic learning rate | 5e-4 (Walker_walk, Cheetah_run, Walker_run) 1e-4 (Other tasks) | Actor learning rate | 5e-4 (Walker_walk, Cheetah_run, Walker_run) 1e-4 (Other tasks) |
| Critic hidden dim | 1024 | Actor hidden dim | 1024 |
| Critic hidden layer | 2 | Actor hidden layer | 2 |
| Critic activation function | ReLU | Actor activation function | ReLU |
| Optimizer | Adam | | |

Table 9: The hyperparameters of SAC

| Hyper-parameter | Value | Hyper-parameter | Value |
|---|---|---|---|
| Size of policy-aligned buffer $N$ | 10 | Data augmentation ratio $\tau$ | 20 |
| Hybrid experience replay sample ratio $\omega$ | 0.5 | Min/Max length of subsampled snippets | [35, 45] |

Table 10: The hyperparameters of QPA

| Hyper-parameter | Value | Hyper-parameter | Value |
|---|---|---|---|
| Unlabeled batch ratio | 4 | Threshold | 0.99 |
| Loss weight | 1 | Min/Max length of cropped segment | [45, 55] |
| Segment length before cropping | 60 | | |

Table 11: The hyperparameters of SURF

| Hyper-parameter | Value | Hyper-parameter | Value |
|---|---|---|---|
| Length of segment | 50 | Unsupervised pre-training steps | 9000 |
| Size of query selection buffer | 100 | | |

Table 12: The hyperparameters of PEBBLE

## I.8 PARAMETER FOR TASKS

Determining the number of feedback instances for each task, the interval between queries, and the quantity of feedback per query can be quite challenging. We have summarized the experimental settings from the QPA (Hu et al., 2023) and SURF (Park et al., 2022) papers in Table 13. The experiments in our paper strictly adhere to the settings outlined in this table.

| Hyper-parameter | Total feedback | Frequency of feedback | Queries number per session |
|---|---|---|---|
| Walker-walk | 100 | 20000 | 10 |
| Walker-run | 100 | 20000 | 10 |
| Cheetah-run | 100 | 20000 | 10 |
| Quadruped-walk | 1000 | 30000 | 100 |
| Quadruped-run | 1000 | 30000 | 100 |
| Humanoid-stand | 10000 | 5000 | 50 |
| Drawer-open | 4000 | 5000 | 20 |
| Sweep-into | 10000 | 5000 | 50 |
| Hammer | 10000 | 5000 | 50 |

Table 13: The hyperparameters of tasks

