# OpenReview forum: "Outward Odyssey: Improving Reward Models with Proximal Policy Exploration for Preference-Based Reinforcement Learning"
_ICLR.cc/2025/Conference — Submitted to ICLR 2025_

### Official Review · Reviewer_7kxy · 2024-10-20

**Soundness:** 1
**Presentation:** 2
**Contribution:** 2
**Rating:** 5
**Confidence:** 5

**Summary:**

This paper focuses on developing a preference-based RL algorithm that increases the data coverage with a preference buffer. More specifically, it is found that the reward model cannot accurately predict rewards for trajectories that are out of distribution from the original training set. Therefore, the authors propose the Proximal Policy Exploration (PPE) algorithm that encourages the agent to explore data that falls outside of the preference data distribution, but close to the agent’s current policy.

**Strengths:**

1. The motivating example in Section 3 helps demonstrate the OOD issue.
2. I think it was interesting to evaluate the reward model by comparing the ground truth return and the learned return via the Spearman correlation coefficient in Section 3.1.
3. Performing OOD Dection seems like a novel addition to the PbRL algorithm.

**Weaknesses:**

Similar to prior work/findings:

1. PPE seems extremely similar to QPA [1]. Both methods seem to be focused on selecting queries that are close to the agent’s current policy. However, the authors only briefly mention this work in the related works. For example, lines 249-251 from this paper state,“We have designed the [PPE] method, to actively encourage the agent to explore data that falls outside of the preference buffer distribution but within the vicinity of the current policy’s distribution”.
Snippet taken from [1] “In particular, it is crucial to ensure that the pairs of segments (σ0, σ1 ) selected for preference queries stay close to the current policy’s visitation distribution”.
The authors need to more clearly outline how their work is different from QPA.
2. The authors’ third contribution is about the reward model’s accuracy on out-of-distribution data. In particular, the reward model can only output accurate values for trajectories it has previously trained on. However, I’m not convinced this is a new finding. The reward models are trained via supervised learning, therefore this seems like an overfitting/generalization issue, which has been heavily studied before.


Lack of Experimental Details:

The authors refer the readers to Appendix G for a complete understanding of the experimental details, but Appendix G only contains hyperparameter details. There is no mention of:

1. How were preferences elicited? I’m assuming preferences were obtained through a scripted teacher but it does not mention it anywhere in the text.

2. How many seeds were run?

3. What are the error bars being visualized in Figures 2 and 3?


Lack of Supporting Evidence for the Effectiveness of the Proposed Algorithm:

1. PPE was evaluated in 9 environments, however, PPE only appeared to have distinctly higher performance in Humanioid-stand. There are significant overlaps in error bars in all other environments.

2. In addition, the authors did not perform statistical analysis on any evaluation metrics such as final performance or area under the curve.

3. Did any experiments involve actual human preferences? The authors note that PPE achieves significant improvement in human feedback efficiency, however, I’m not sure if any humans were involved. I don’t think any claims can be made about human teachers if only simulated/scripted teachers were used.

4. Unclear why PPE was integrated with QPA. See Question 1.

Minor comments:

The authors note “We selected six distinct complex tasks from DMControl”, however, I would argue that (Walker-walk and Walker-run), and (Quadruped-walk and Quadruped-run) are not distinct. I think this claim is a bit too strong.


[1] QPA: QUERY-POLICY MISALIGNMENT IN PREFERENCE BASED REINFORCEMENT LEARNING
https://arxiv.org/pdf/2305.17400

**Questions:**

1. Why is PPE being integrated with QPA and not PEBBLE? Both RUNE and SURF are integrated on top of PEBBLE. Therefore, it seems like an unfair advantage to PPE if it is being added on top of a more advanced PbRL algorithm. Or put differently, why did the authors not integrate the other PbRL baselines on top of QPA? In the QPA paper, the authors note that it can be integrated on top of any off-policy PbRL algorithm, including SURF and RUNE.

2. Does PPE improve performance if it is added on top of other PbRL algorithms, as the authors mention it is highly compatible with existing strategies and frameworks.

---

> ### Author Response · Authors · 2024-11-14
> **Reply to Reviewer 7kxy - Part 1**
>
> We sincerely appreciate your valuable insights and the diligent effort you have invested in the review. In response to the concerns you have raised, please find our replies below:
>
> >**W1.** Similar to prior work
>
> >>**1).** PPE seems similar to QPA
>
> **A:** The QPA method addresses the issue of 'Query-Policy Misalignment' by adjusting the query selection method to align with the current policy's distribution.
>
> In contrast, PPE focuses not on the query selection phase but on the exploration phase during the interaction between the policy and the environment. PPE identifies that the reward model's reliability is compromised when the coverage of the preference buffer is insufficient. Therefore, PPE aims to enhance data coverage used for training the reward model, ensuring more reliable results.
>
> The issue that PPE addresses is also present in QPA, as QPA does not actively expand the data coverage for training the reward model, according to the paper.
>
> Since the query phase and data exploration are implemented at different stages, we believe that the motivations and implementations of these two works are fundamentally different.
>
> >> **2).** Problem has been heavily studied before
>
> **A:** Fortunately, we have not found research similar to ours in previous online PBRL papers. We agree with you that ‘reward models are trained via supervised learning, therefore this seems like an overfitting/generalization issue’, but we would like to clarify that the impact of this issue in online Pbrl settings has not been thoroughly considered or discussed in prior work.
>
> Moreover, there is no established solution available for reference. We would appreciate it if you could point out any relevant papers, and we will actively engage in discussing them.
>
> > **W2.** Lack of Experimental Details
>
> Thank you for pointing out this issue. We will include the relevant content in the revised version of the paper.
>
> >>**1).** How were preferences elicited?
>
> **A:** We used the same approach as PEBBLE, SURF, RUNE, and QPA, utilizing the B-pref framework [1] to script access to the ground truth reward, thereby simulating human preference labels.
>
> [1]  *Lee, Kimin, et al. "B-pref: Benchmarking preference-based reinforcement learning." arXiv preprint arXiv:2111.03026 (2021).*
>
> >>**2).** How many seeds were run?
>
> **A:** 5 seed for each experiment.
>
> >>**3).** What are the error bars being visualized in Figures 2 and 3?
>
> **A:** The error bars in Figures 2 and 3 represent the 95% confidence intervals derived from the experimental results of methods using five different seeds.
>
> > **W3.** Lack of Supporting Evidence
>
> >> **1).** About performance
>
> **A:** We observed that the mean performance of the policy improved noticeably with the use of PPE. As for the "overlaps in error bars," this is indeed challenging to avoid. PBRL is inherently influenced by the preference buffer, which is generated online, resulting in significant variance in the results. This issue is also common in prior works such as SURF, PEBBLE, RUNE, and QPA.
>
> >>**2).** In addition, the authors did not perform statistical analysis on any evaluation metrics such as final performance or area under the curve
>
> **A:** You can refer to Appendix D, where we have documented the final performance and the variance of the experimental results.
>
> >>**3).** About actual human preferences
>
> **A:** Thank you for your insightful comment. You are absolutely correct, and we will replace "human feedback" with "preference feedback" in the text.
>
> >**W4.** Minor comments
>
> **A:** Thank you for your feedback. We will remove the word "distinct" from the text.

---

> ### Author Response · Authors · 2024-11-14
> **Reply to Reviewer 7kxy - Part 2**
>
> >**Q1.** Why is PPE being integrated with QPA and not PEBBLE?
>
> **A:** Firstly, QPA is the latest state-of-the-art algorithm, and we wanted to test whether our method could provide further improvements on top of it. Our experimental results demonstrate that our approach indeed achieves this.
>
> Secondly, when the goal is to enhance the current policy, a reward model that evaluates more accurately around the current policy is undoubtedly a better choice than one that is globally better. QPA naturally reduces the number of queryable trajectory pairs $(\tau_0, \tau_1)$, as referenced by the "policy-aligned buffer size of *N*=10" in Section 6 of [1]. Therefore, integrating PPE with QPA allows for a more reliable reward model when evaluating the trajectories around the current policy.
>
> This is why we integrated PPE with QPA.
>
> It's also worth noting that in Appendix C.1 of [1], specifically in Algorithm 1, Lines 8-9, QPA already incorporates the SURF trick in its implementation. Therefore, we focused on evaluating the effectiveness of PPE with RUNE.
>
> [1] *QPA: QUERY-POLICY MISALIGNMENT IN PREFERENCE BASED REINFORCEMENT LEARNING https://arxiv.org/pdf/2305.17400*
>
> >**Q2.** PPE added on PEBBLE.
>
> **A:** Thank you for highlighting this issue. We agree that the relevant validation is important. The table below presents our validation results, demonstrating that integrating PPE with PEBBLE leads to performance improvements. We also highlight the differences in impact between PPE and RUNE's exploration strategies when applied to PEBBLE.
>
> To convert your LaTeX table into a format suitable for OpenReview, you need to ensure that it is compatible with the Markdown-like syntax used by OpenReview. OpenReview supports a limited subset of Markdown for tables, so you will need to simplify the formatting and remove any LaTeX-specific commands. Here's how you can represent your table in a format that OpenReview can display:
>
> | Task                | PEBBLE           | PEBBLE+RUNE      | PEBBLE+PPE       |
> |---------------------|------------------|------------------|------------------|
> | Walker-walk\_1e2    | 453.43 ± 159.43  | 414.62 ± 182.16  | **499.73 ± 82.75** |
> | Walker-run\_1e2     | 188.21 ± 79.86   | 251.48 ± 104.98  | **257.64 ± 58.59** |
> | Quadruped-walk\_1e3 | 369.51 ± 134.22  | 440.30 ± 296.02  | **451.06 ± 223.27** |
> | Quadruped-run\_1e3  | 314.91 ± 120.87  | 231.85 ± 60.14   | **373.09 ± 149.10** |
> | Cheetah-run\_1e2    | 545.77 ± 130.00  | 508.60 ± 186.06  | **569.54 ± 84.27**  |
> | Humanoid-stand\_1e4 | 306.08 ± 171.92  | 351.10 ± 197.75  | **357.13 ± 76.15**  |
>
>
> Furthermore, as we mentioned earlier in Section 3.1, Figure 1(b), and lines 179-181 of the text, RUNE's approach, which is based on the variance of the reward model output, does not effectively identify data that is out-of-preference distribution. PPE, on the other hand, aims to explore more out-of-distribution (OOD) data. This differs from RUNE, which uses an additional extrinsic reward to guide policy exploration, representing two distinct phases. PPE primarily influences exploration by introducing data around the policy that is out-of-preference distribution through the target policy $\pi_T$, whereas RUNE's extrinsic reward alters the target policy $\pi_T$ itself.

---

> > ### Comment · Reviewer_7kxy · 2024-11-26
> > **Response to Authors**
> >
> > I appreciate the additional experiments the authors included. I better understand the difference between QPA and PPE. But two of my main concerns are still valid.
> >
> > - In most of the experiments, PPE does not appear to perform statistically better than the other baselines. PPE appears to be yield comparable performance. There are overlapping confidence bars in most of the figures, which does not necessarily rule out statistical significance. But it would be helpful if the authors performed some statistical tests as I previously mentioned, such as a Welsh T-test, etc. This is a useful resource [1].
> > - This is a preference learning algorithm, which is supposed to learn reward functions from human preferences. However, there are no human subject studies, not even a small pilot study. This makes it difficult to determine the validity of this approach.
> >
> > Overall, this is an empirical paper, and I believe more time needs to be spent revising this paper (improving the experiments in particular) before it is ready for acceptance.
> >
> > [1] https://arxiv.org/abs/1904.06979

---

> ### Author Response · Authors · 2024-11-21
> **Seeking Your Further Insights and Feedback**
>
> Dear Reviewer 7kxy,
>
> We hope this message finds you well. It has been **eight days** since we provided our responses to your suggestions, and we have not yet received a reply. **We sincerely hope that our responses have addressed your concerns and would appreciate the opportunity to engage in further discussion**. We kindly encourage your continued involvement in this process.
>
> Thank you for your time and consideration.

---

> ### Author Response · Authors · 2024-11-23
> **Looking Forward to Further Discussion**
>
> Dear Reviewer 7kxy,
>
> We sincerely appreciate your valuable time and effort in reviewing our paper. We hope that our responses have adequately addressed all your questions and concerns.
>
> If there are any remaining issues or if you require further information, please do not hesitate to let us know. We are more than happy to provide additional materials or discuss any aspect of our work in more detail.
>
> Thank you once again for your consideration.
>
> Best regards,
>
> The Authors

---

> ### Author Response · Authors · 2024-11-30
> **Discussion with Reviewer 7kxy - Part 1**
>
> We greatly appreciate your suggestion and hope that the following explanation helps address any remaining concerns you may have.
>
> > **1.** It would be helpful if the authors performed some statistical tests as I previously mentioned, such as a Welsh T-test, etc.
>
> **A:** Thank you for your suggestions. We conducted statistical tests on all the experimental results.
>
> The results of the Welsh T-test comparing the final performance of PPE with that of other baselines：
>
> |  | PEBBLE | SURF | RUNE | QPA |
> | --- | --- | --- | --- | --- |
> | walker_walk | 6.0247 | 5.1586 | 5.7963 | 1.5858 |
> | Walker-run | 5.7399 | 4.0779 | 4.1103 | 0.9492 |
> | Quadruped-walk | 2.9398 | 1.1491 | 1.4278 | 0.6062 |
> | Quadruped-run | 1.5780 | 2.1105 | 3.4360 |  0.6763 |
> | Cheetah-run | 1.6605 | 2.9569 | 1.6168 | 1.0810 |
> | Humanoid-stand | 3.4696 | 12.0554 | 2.5250 | 6.7143 |
> | Drawer-open | 1.7871 | 0.9496 | 0.7384 | 0.9496 |
> | Sweep-into | 1.3052 | 2.4492 | -0.8298 | 1.2485 |
> | Hammer | 2.2834 | 0.9249 | 0.5327 | 0.8834 |
>
> The results of the Welsh T-test comparing the performance of PPE with that of other baselines across all evaluations during training:
>
> |  | PEBBLE | SURF | RUNE | QPA |
> | --- | --- | --- | --- | --- |
> | walker_walk | 17.2363 | 7.7909 | 18.6282 | 4.4137 |
> | Walker-run | 31.8804 | 23.5488 | 26.6891 | 5.9516 |
> | Quadruped-walk | 14.4218 | 6.5593 | 11.7086 | 1.0420 |
> | Quadruped-run | 13.1904 | 16.8964 | 20.4076 |  4.5345 |
> | Cheetah-run | 11.5897 | 6.9901 | 8.7536 | 5.8251 |
> | Humanoid-stand | 26.5792 | 24.8984 | 23.5906 | 7.9869 |
> | Drawer-open | 10.4972 | 2.0464 | -1.3821 | 2.6691 |
> | Sweep-into | 7.0637 | 1.1292 | -5.8169 | 2.1000 |
> | Hammer | 16.4993 | 8.7253 | 1.0280 | 2.4130 |
>
> The positive value indicates the degree of improvement of PPE over the corresponding baseline method. Overall, these results can prove that using the PPE method can further enhance algorithm performance.
>
> > **2.** This is a preference learning algorithm, which is supposed to learn reward functions from human preferences. However, there are no human subject studies, not even a small pilot study. This makes it difficult to determine the validity of this approach.
>
> **A:** Thank you for your suggestion, we find it very valuable.
>
> >> **Implementation**
>
>  In response, we have conducted relevant experiments. The detailed implementation and screenshots of the interaction interface used to obtain preference feedback from humans have been updated in the revised version's Appendix I.6.
>
> We implemented the human interaction pipeline using the following code:
>
>
>
> ```
> def get_label(self, sa_t_1, sa_t_2, physics_seg1, physics_seg2):
>     frame_height, frame_width, channels = physics_seg1[0,0].shape
>
>     # Create a video writer
>     output_width = frame_width * 2  # The merged width is twice the original.
>     output_height = frame_height
>     fps = 30  # Set the frame rate.
>
>     # Save video
>     human_labels = np.zeros(sa_t_1.shape[0])
>     for seg_index in range(physics_seg1.shape[0]):
>         # Render the pairs of segments and save the video
>         # Create a video writer using imageio
>         with iio.get_writer(f'output.mp4', fps=fps) as writer:
>             # Iterate over all frames.
>             for frame0, frame1 in zip(physics_seg1[seg_index], physics_seg2[seg_index]):
>                 # Horizontally merge frames
>                 combined_frame = np.hstack((frame0, frame1))
>                 # Write to the video file
>                 writer.append_data(combined_frame)
>         labeling = True
>         # Provide labeling instruction and query human for preferences
>         while labeling:
>             print("\n")
>             print("---------------------------------------------------")
>             print("Feedback number:", seg_index)
>             # Preference:
>             # 0: segment 0 is better
>             # 1: segment 1 is better
>             while True:
>                 # Check if it is 0/1/number type preference
>                 try:
>                     rational_label = input("Preference: 0 or 1 or other number")
>                     rational_label = int(rational_label)
>                     break
>                 except ValueError:
>                     print("Wrong label type. Please enter 0/1/other number.")
>             print("---------------------------------------------------")
>             human_labels[seg_index] = rational_label
>             labeling = False
>     # Remove the hard-to-judge pairs of segments
>     cancel = np.where((human_labels != 0) & (human_labels != 1))[0]
>     human_labels = np.delete(human_labels, cancel, axis=0)
>     sa_t_1 = np.delete(sa_t_1, cancel, axis=0)
>     sa_t_2 = np.delete(sa_t_2, cancel, axis=0)
>     print("Valid query number:", len(human_labels))
>     return sa_t_1, sa_t_2, human_labels.reshape(-1, 1)
>
> ```
>
> In simple terms, we store the two behavior segments that need to be compared in a single video file, allowing humans to provide preference feedback by observing this video.

---

> > ### Comment · Reviewer_7kxy · 2024-12-01
> > **Statistical Testing**
> >
> > What values are being shown in the tables? Can you provide the p-value for the corresponding statistical tests? If the p-value is less than the predefined significance level (commonly 0.05), then this would indicate statistical significance.

---

> > > ### Author Response · Authors · 2024-12-02
> > > **Discussion with Reviewer 7kxy - Part 3**
> > >
> > > Thank you very much for your prompt response. In response to your questions, we provide the following explanation:
> > >
> > > >**1.** What values are being shown in the tables?
> > >
> > > **A:** The table in Part 1 presents the t-statistic from Welch's T-test, which we use to measure the size of the difference between two sample means relative to the variability in the samples. A larger t-statistic indicates a more significant difference between the means.
> > >
> > > The table in Part 2 shows the final performance of agents trained using human feedback, evaluated under the hand-engineered rewards provided by DMControl. We also conducted a statistical analysis, specifically a Welch's T-test between QPA+PPE and QPA, yielding a t-statistic of 1.8515 and a p-value of 0.0349.
> > >
> > > > **2.** Can you provide the p-value for the corresponding statistical tests?
> > >
> > > We have compiled the p-values for the Welch's T-test comparing the performance of PPE with other baselines across all evaluations during training in the table below:
> > >
> > > | **Task/Algorithm** | **PEBBLE** | **SURF** | **RUNE** | **QPA** |
> > > | --- | --- | --- | --- | --- |
> > > | Walker_walk | <0.001 | <0.001 | <0.001 | <0.001 |
> > > | Walker_run | <0.001 | <0.001 | <0.001 | <0.001 |
> > > | Quadruped-walk | <0.001 | <0.001 | <0.001 | 0.2977 |
> > > | Quadruped-run | <0.001 | <0.001 | <0.001 | <0.001 |
> > > | Cheetah-run | <0.001 | <0.001 | <0.001 | <0.001 |
> > > | Humanoid-stand | <0.001 | <0.001 | <0.001 | <0.001 |
> > > | Drawer-open | <0.001 | 0.0412 | 0.1675 | 0.0078 |
> > > | Hammer | <0.001 | <0.001 | 0.3041 | 0.0160 |
> > > | Swep-into | <0.001 | <0.001 | 0.0184 | <0.001 |
> > >
> > > For most tasks, the algorithms PEBBLE, SURF, RUNE, and QPA show significant differences from PPE (p-value < 0.05), indicating that these algorithms perform significantly differently from PPE on these tasks.
> > >
> > > In three cases, the p-value is greater than 0.05, such as for RUNE in the Drawer-open task, RUNE in the Hammer task, and QPA in the Quadruped-walk task. Although there is no significant difference, PPE remains one of the best methods.
> > >
> > > **Thank you for your valuable feedback. We will thoughtfully incorporate your suggestions into our revised version. We hope we have addressed your concerns, but if you have any further questions or concerns, please do not hesitate to let us know. Your input is incredibly important to us and essential for enhancing the quality of our research.**

---

> > > > ### Comment · Reviewer_7kxy · 2024-12-02
> > > >
> > > > I appreciate your quick response to adding the p-values.
> > > >
> > > > These results do seem surprising, given that the confidence intervals in Figures 2 and 3 from the main text seem to greatly overlap. It would be clearer if all of the plots and tables reported the same measure of spread (i.e., standard deviation, variance, or confidence interval). For example, the authors reported using 95% CI for Figures 2 and 3 (main text), but in Table 1 of the appendix variance is reported, and then in the table discussing the human feedback experiment, standard deviation is reported.

---

> > > > > ### Author Response · Authors · 2024-12-02
> > > > > **Discussion with Reviewer 7kxy - Part 4**
> > > > >
> > > > > We completely agree with you that using a consistent measure of spread is crucial for the clarity of results, and we will carefully implement this suggestion in the revised version. We will also include additional experimental results from the rebuttal phase in the main text, while ensuring consistency in the measures of spread. Furthermore, we will incorporate p-values to enhance the reliability of our reported results.
> > > > >
> > > > > Thank you for your valuable suggestions during the discussion phase. We sincerely hope that our responses have addressed some of your concerns. If they have, we would greatly appreciate any reconsideration of the score. Thank you once again.

---

> > > > > > ### Comment · Reviewer_7kxy · 2024-12-02
> > > > > >
> > > > > > I will increase my score to a 5 because I acknowledge that the authors put in a considerable amount of work during the rebuttal period to improve the paper.
> > > > > >
> > > > > > But overall I still maintain that this work is not ready for acceptance, and could benefit from more time. I think the clarity and presentation of the results need to be improved. In addition, although the authors added an experiment with human feedback, it was only 2 users, making it difficult to make any claims.

---

> > > > > > > ### Author Response · Authors · 2024-12-04
> > > > > > > **Discussion with Reviewer 7kxy - Part 5**
> > > > > > >
> > > > > > > Thank you for your suggestions and for increasing the score.
> > > > > > >
> > > > > > > > **Clarity**
> > > > > > >
> > > > > > > **A:** With the help of the reviewers during the discussion phase, we have enhanced the readability of the paper by correcting minor misstatements, adding experimental details, providing additional explanations of the experimental data, and including numerous additional experiments. We believe these improvements significantly aid readers in better understanding our work. **As per your suggestion, we have standardized the use of standard deviation as shading in our figures, which will be updated in future versions.**
> > > > > > >
> > > > > > > > **More Volunteers for Human Feedback**
> > > > > > >
> > > > > > > **A:** Our experiment was inspired by your question: "*This is a preference learning algorithm, which is supposed to learn reward functions from human preferences. However, there are no human subject studies, not even a small pilot study. This makes it difficult to determine **the validity of this approach**.*" We have provided video evidence in the anonymous link: https://anonymous.4open.science/r/Demos-ICLRDiscussion, which adequately demonstrates **the validity of this approach**.
> > > > > > >
> > > > > > > Similar to the QPA paper, we used two humans for preference labeling. In studies like SURF and RUNE, human preferences were not used due to the challenge of evaluating algorithms quantitatively and quickly, as noted in [1], Page 5. In PEBBLE, multiple volunteers were not recruited; instead, frame-by-frame demonstrations were provided. Even with more humans involved, evaluations under hand-engineered rewards would only **serve as a reference**. The actual human feedback experimental effects can only be assessed by directly observing the performance of the trained agent, as shown in the video provided in the anonymous link: https://anonymous.4open.science/r/Demos-ICLRDiscussion. We hope this clarifies your concerns.
> > > > > > >
> > > > > > > We hope you will consider our perspective carefully, and we would be grateful if you could acknowledge our efforts with an increased score.
> > > > > > >
> > > > > > > [1] Park J, Seo Y, Shin J, et al. SURF: Semi-supervised reward learning with data augmentation for feedback-efficient preference-based reinforcement learning. arXiv preprint arXiv:2203.10050, 2022.

---

> ### Author Response · Authors · 2024-11-30
> **Discussion with Reviewer 7kxy - Part 2**
>
> >> **Experiment Setup**
>
> We recruited two volunteers to conduct experiments using QPA+PPE and QPA on the cheetah-run task. Each volunteer was required to label 100 preference feedbacks, with 10 preference feedbacks per query, consistent with the task setup described in Appendix I.8.
>
> >> **Experiment Results**
>
> The final performance results of the experiments are as follows:
>
> | **Algorithm** | **Mean** | **Std Dev** |
> | --- | --- | --- |
> | QPA+PPE | 343.97 | 132.78 |
> | QPA | 289.03 | 83.24 |
>
> We conducted statistical tests on all evaluation results during the training process:
>
> Welch's T-test between QPA+PPE and QPA: t-statistic = 1.8515
>
> >> **Analysis**
>
> This indicates that, based on the hand-engineered rewards provided by DMControl, QPA+PPE performs better than QPA. However, hand-engineered results are only a reference indicator. As noted in the footnote at the bottom of Page 5 in the SURF paper [1], **"While utilizing preferences from the human teacher is ideal, this makes it hard to evaluate algorithms quantitatively and quickly."**
> This might explain why some previous papers, such as SURF [1] and RUNE [2], did not include experiments using human feedback. Human preferences are more difficult to quantify compared to preferences generated from ground truth rewards via scripts, making it challenging to provide effective metrics to evaluate algorithms quantitatively and quickly.
>
> To more clearly observe the differences between agents trained with QPA+PPE and QPA under human feedback settings, we have uploaded videos of the final agent performance and the  videos provided to humans during a query by different algorithms to the following **anonymous link**:
>
> [https://anonymous.4open.science/r/Demos-ICLRDiscussion](https://anonymous.4open.science/r/Demos-ICLRDiscussion/query_ppe2.mp4)
>
> We found that the policy learned by the QPA+PPE method aligns more closely with human understanding of how animals like cheetahs run, whereas the policy learned by QPA shows greater variability.
>
> By observing the 'query phase' interaction videos with humans, we noticed that the behavior segment pairs provided by QPA to humans showed less variability and tended to be similar. In contrast, QPA+PPE, by actively exploring near-policy OOD data, increased the diversity of candidate behavior segments (in $\mathcal{D}^{cp}$), resulting in a higher likelihood of presenting humans with more different behaviors during the 'query phase'.
>
> When the diversity of candidate behavior segments is greater, humans are more likely to select behaviors that align with their understanding. Consequently, the behavior patterns of agents trained with QPA+PPE more closely match human perceptions of animal locomotion compared to those trained with QPA.
>
> [1] *Park J, Seo Y, Shin J, et al. SURF: Semi-supervised reward learning with data augmentation for feedback-efficient preference-based reinforcement learning[J]. arXiv preprint arXiv:2203.10050, 2022.*
>
> [2] *Liang X, Shu K, Lee K, et al. Reward uncertainty for exploration in preference-based reinforcement learning[J]. arXiv preprint arXiv:2205.12401, 2022.*

---

### Official Review · Reviewer_KyBu · 2024-10-29

**Soundness:** 3
**Presentation:** 3
**Contribution:** 3
**Rating:** 6
**Confidence:** 3

**Summary:**

The authors introduce a novel PbRL method, that is based on the idea, that it is important to increase the OOD error of the reward model. Therefore, they introduce a query method that considers a mixture of OOD and ID samples. Futhermore, they employ a Morse network for obtaining well calibrated uncertainty estimates that allow for efficient OOD detection. The algorithm is evaluated on several locomation and robotics benchmark tasks and compared to 4 existing PbRL algorithm. The authors also included a ablation study wrt. two of the hyper parameters and the distinct modules of their algorithm.

**Strengths:**

The authors try to tackle a known problem with a new perspective. It is known, that PbRL is subject to a dual exploration problem: The policy space and the reward space. However, exploring the reward space via an explicit OOD detection is novel, as far as the reviewer knows. Furthermore, they utilize novel Morse networks, not applied in PbRL before. With a good benchmark set of test domains and algorithms, they establish the usefulness of the approach. Resultingly originality and impact are good. However, significance is limited due to clarity issues and some remaining research questions.

**Weaknesses:**

The most impactful weakness of the paper are issues of clarity. It seems, that the authors are not using a form of reward-based PbRL, but a variant of direct policy learning. (see A Survey of Preference-Based Reinforcement Learning Methods, Wirth, 2017) $M_\phi$ approximates an action distribution, according to Eq. 3, not a reward distribution, which is then used to modify a policy $\pi_T$ (Eq.6). However, the origin of $\pi_T$ is never discussed. Section 2 indicates that this may be a reward-based PbRL policy, but this is not clear. In case it is, this should be clarified and also explained which method is used to derive a policy from the learned reward. The according updates should be added to Alg.2. These issues also impacts the claim "Our method, as outlined in Section 3.2, is designed to be orthogonal and highly compatible with existing strategies", because the orthogonality is not visible. These clarity issues are probably not difficult to resolve, but are quite impactful. There are also some clarity issues wrt. the motivating example (see questions).

A second problem is, that this is difficult to attribute the performance gains to the coverage improvement. The coverage idea implicitly assumes that all parts of the trajectory space are equally important, which is usually not true. It is sufficient to obtain a reward function that induces the same optimal policy as the true reward and guides the policy learning process towards that policy. Therefore, methods combining expected reward and uncertainty are usually used (like RUNE). That it is better to "only" consider coverage would be a very interesting insight, but the algorithm deviates from conventional PbRL methods in two other aspects: The preference exploration is defined as a policy, not a reward scheme and the uncertainties are modelled using an RBF-kernel based method. Kernel methods are known to be better wrt. ODD uncertainty (as compared to e.g. ensembles, like used in RUNE). Therefore, the ablation studies using ensembles instead of Morse or defining the coverage bonus in the reward space are of major interest. However, the problem of unclear effect attribution is not substantial enough to prevent acceptance, as the benefit of the full method is sufficiently established. Although, it would greatly improve the contribution.

A further improvement can be achieved by a bit more extensive discussion of Morse networks, as they should not be considered an established method (there seems to be only an arxiv short paper, nothing peer reviewed). Foremost is the question, how scalable Morse networks are. Most PbRL methods abstain from using Kernel-based approaches, despite their advantages, due to the costly scaling wrt. number training data points.

Lastly, the structure of the paper could be improved a bit moving the related work discussion in the introduction to the related work section and the OOD detection from preliminaries to method.

**Questions:**

- Preliminaries: $y_p$ is not defined, but seems to be restricted to strict preference?
- Motivation Example: Are region 1-9 sizes (as indicated by line 171) or an index?
- Motivating Example: What is the is the evaluation region for used for 1d?
- This implies that $\hat{a} = a$ only when the pair (s, a) originates from the preference buffer. This claim depend on the form of $f$, e.g. a linear approximator may not ensure this. Is this ensured by the Morse network?
- Some baseline results deviate from reported literature (e.g. SURF, QPA drawer-open). Is the statement "we also made use of the official code repositories" true for all baselines or are some re-implemented?
- The method seems quite sensitive against hyperparameter changes (4.2). Is there a set of reasonable hyperparameters for unseen tasks?

---

> ### Author Response · Authors · 2024-11-15
> **Reply to Reviewer KyBu - Part 1**
>
> We greatly value your insightful feedback and the thorough review you have provided. In response to the concerns you've highlighted, please find our detailed replies below:
>
> >**W1.** Issues of clarity
>
> Thank you very much for your feedback regarding the clarity of our paper. We apologize for any confusion caused. We will incorporate the following clarifications in the revised version of the paper to enhance its overall clarity.
>
> >>**1).** If using a form of reward-based PbRL?
>
> **A:** Yes, in the implementation of our code, we use QPA as the backbone, which is a state-of-the-art reward-based PbRL method. Our paper focuses on enhancing policy exploration by using data that is both close to the current policy and has greater coverage to train the reward model.
>
> >>**2).** About the origin of $\pi_T$
>
> **A:** The policy $\pi_T$ is derived by training a SAC agent using the learned reward model, as described in line 17 of Algorithm 2. This approach is consistent with the frameworks of methods like PEBBLE, SURF, and QPA. To improve clarity and reduce potential misunderstandings, we plan to revise line 17 of Algorithm 2 to read: "*Optimize $\pi_T$ via the SAC method*".
>
> >>**3).** About orthogonality
>
> First, we conduct additional experiments by applying PPE to PEBBLE to demonstrate that PPE can be integrated with other methods to achieve performance improvements. This content will be updated in Appendix D of the revised version.
>
> | Task                | PEBBLE           | PEBBLE+RUNE      | PEBBLE+PPE       |
> |---------------------|------------------|------------------|------------------|
> | Walker-walk\_1e2    | 453.43 ± 159.43  | 414.62 ± 182.16  | **499.73 ± 82.75** |
> | Walker-run\_1e2     | 188.21 ± 79.86   | 251.48 ± 104.98  | **257.64 ± 58.59** |
> | Quadruped-walk\_1e3 | 369.51 ± 134.22  | 440.30 ± 296.02  | **451.06 ± 223.27** |
> | Quadruped-run\_1e3  | 314.91 ± 120.87  | 231.85 ± 60.14   | **373.09 ± 149.10** |
> | Cheetah-run\_1e2    | 545.77 ± 130.00  | 508.60 ± 186.06  | **569.54 ± 84.27**  |
> | Humanoid-stand\_1e4 | 306.08 ± 171.92  | 351.10 ± 197.75  | **357.13 ± 76.15**  |
>
> From a theoretical perspective, our method focuses on **proximal policy** exploration during the data collection phase, specifically in the interaction stage between $\pi_E$ and the environment. Current pbrl methods do not address improvements in this particular phase.
>
> PEBBLE serves as a foundational PbRL framework. SURF enhances the reward learning process through semi-supervised use of unlabeled data. RUNE modifies the optimization direction of $\pi_T$ by introducing intrinsic rewards, and QPA optimizes the query selection process. Each of these methods targets different stages of the PbRL process compared to our approach.
>
> More specifically, the typical online PbRL process consists of: **data collection** (PPE) → **data selection and preference labeling**  (QPA) → **learning the reward model using preference labels $(\tau_0,\tau_1,y_p)$** (SURF, PEBBLE) → **optimizing $\pi_T$ with the learned reward model via reinforcement learning methods** (RUNE, QPA).
>
> Therefore, we believe that our method is theoretically compatible with these approaches, which is why we state that our method is "highly compatible with existing strategies".

---

> ### Author Response · Authors · 2024-11-15
> **Reply to Reviewer KyBu - Part 2**
>
> >**W2.** It is difficult to attribute the performance gains to the coverage improvement
>
> >>**1).** Is that all parts of the trajectory space are equally important?
>
> **A:** We apologize for any misunderstanding. Like you, we **do not agree** with the notion that "all parts of the trajectory space are equally important".
>
> The core focus of our paper is to enhance the coverage of the data distribution for **proximal policies**. As illustrated by the phenomenon in the toy example in Section 3.1, **training the reward model with data that has greater coverage of the proximal policy's data distribution results in a more reliable evaluation capability around the current policy.** This, in turn, leads to a more reliable optimization direction for $\pi_T$ guided by the reward model.
>
> Since our backbone is QPA, during the "data selection and preference labeling" phase, we are inherently limited to selecting data generated by relatively proximal policies. Therefore, we only need to further promote the collection of data from the out-of-preference data distribution to achieve our goal of training the **reward model** with data that has **greater coverage of the proximal policy's data distribution.**
>
> Thus, our paper emphasizes not only **coverage** but also **proximal policy**, forming a complete PPE approach. This does not preclude demonstrating that PPE is also effective on PEBBLE. In the code released by B-pref [1][2], the main difference between QPA and PEBBLE lies in the '*max_size*' parameter setting in the '*RewardModel*' class. This parameter represents the number of episodic trajectories stored closest to the current time, serving the data pool for the "data selection and preference labeling" phase, following a first-in-first-out storage mode. In PEBBLE, it is set to 100, while in QPA, it is set to 10. Therefore, during the "data selection and preference labeling" phase, the data for query selection in QPA is inherently nearer to the current policy compared to PEBBLE. Consequently, while PPE does have some effect on PEBBLE, the improvement is not as significant as when applied to QPA.
>
> [1]*Lee, Kimin, et al. B-pref: Benchmarking preference-based reinforcement learning. arXiv preprint arXiv:2111.03026 (2021).*
>
> [2] *https://github.com/rll-research/BPref*
>
> >>**2).** Why not expected reward like RUNE？
>
> **A:** RUNE constructs an intrinsic reward using the variance output from an ensemble of reward models, guiding $\pi_T$ to explore during the policy optimization phase through the introduction of this intrinsic reward.
>
> In Section 3.1, Figure 1.b, we conducted experiments to test whether the variance of the reward model's output could determine if data is OOD, and the results were negative.
>
> Our objectives are: 1. "*to acquire data from the out-of-preference buffer distribution surrounding the proximal policy*" and 2. "*to incorporate this data into the preference buffer for reward model learning*". Therefore, we need an efficient way to collect data.
>
> If we use the OOD metric as an intrinsic reward and influence policy updates in the manner of RUNE, it would be inefficient and indirect. This is because the preference buffer periodically introduces new data, causing the OOD metric to change accordingly.
>
> Using intrinsic rewards is clearly less efficient than our proposed method of directly adjusting the current policy $\pi_T$ as $\pi_E$ to effectively collect data from the "out-of-preference buffer distribution surrounding the proximal policy".
>
> Therefore, I believe that compared to the intrinsic reward approach, our PPE method enables more efficient OOD data collection.
>
> >>**3).** More ablation on OOD detection
>
> **A:** Thank you for your insightful suggestion. We find your idea both intriguing and meaningful. We will consider incorporating these validations to further enrich our experimental conclusions.

---

> ### Author Response · Authors · 2024-11-15
> **Reply to Reviewer KyBu - Part 3**
>
> >**W3.** About morse
>
> >>**1).** More extensive discussion of Morse networks
>
> **A:** In our study, the form of the Morse network we employ is quite straightforward. Compared to using random networks [1] or ensemble networks for OOD detection, the final training form of the Morse network, as shown in Eq. (4), involves an additional step of distancing from OOD samples, which is somewhat akin to contrastive learning. In our research, this approach appears to be relatively reasonable.
>
> [1] *Burda, Yuri, et al. "Exploration by random network distillation." arXiv preprint arXiv:1810.12894 (2018).*
>
> >>**2).** How scalable Morse networks are?
>
> **A:** In our study, we utilized a neural network with a $3\times256$ architecture to learn the function $f_\phi$ required for the Morse network, as described in Eq. (4). We will include this detail as an additional supplement in the revised version of our paper.
>
> Given that our dataset is not very large, especially when QPA is used as the backbone with a dataset size of only '*10 × episode_length*', the precision requirements are not very high. The network primarily needs to provide $\pi_E$ with a direction for exploration. Therefore, it suffices for the network to roughly identify which direction contains more OOD data, which is adequate for the purposes of our paper.
>
> >**W4.** Change the structure
>
> **A:** We greatly appreciate this feedback and will incorporate the suggested structural changes in the revised version of our paper.

---

> ### Author Response · Authors · 2024-11-15
> **Reply to Reviewer KyBu - Part 4**
>
> >**Q1.** $y_p$ is not defined.
>
> **A:** We appreciate this observation. To enhance the clarity of our paper, the revised version will include the following definition in Section 2: “*Given a pair of segments $(\tau^0,\tau^1)$, the teacher gives their preference feedback signal $y_p$ among these segments*”.
>
> >**Q2.** Motivation Example: Are region 1-9 sizes (as indicated by line 171) or an index?
>
> **A:** Thank you for your observation. In Section 3.1, "region 1-9" refers to square regions depicted in Figure 1.a, with the lower-left corner as the origin. The grid is labeled from 0 to 9 on both the horizontal and vertical axes, increasing from left to right and from bottom to top, respectively. For example, "region 3" denotes a $3\times 3$ grid area bounded by the segments from 0 to 3 on both axes.
>
> We will include this clarification in the appendix to minimize any potential misunderstandings for readers.
>
> >**Q3.** Motivating Example: What is the is the evaluation region for used for 1d?
>
> **A:** We apologize for any confusion caused. In the experiment corresponding to Figure 1.d, the evaluation region used is the **same** as the training region. This figure is intended to explore how varying the amount of preference feedback affects the performance of the reward model when both the evaluation and training regions are fixed.
>
> We will include this clarification in the appendix to minimize potential misunderstandings for readers.
>
> >**Q4.** About the ability of Morse network.
>
> **A:** Thank you for your question regarding the capability of the Morse network.
>
> Firstly, as mentioned in W3.2, we employ a $3\times 256$ neural network to learn the function. $f_\phi$ required for the Morse network, using the loss defined in Eq. (4). This approach ensures that the expressiveness of $f_\phi$ surpasses that of a linear approximator.
>
> Secondly, we do not rely on the specific outputs of the Morse network to determine whether data is out-of-distribution (OOD). Instead, we only utilize the gradien $\nabla_a M_{\phi}(s,a)$ and use it as a basis for sampling data in the 'Mixture Distribution Query'. These applications do not demand high precision in the Morse network's outputs; they only require a relative distinction in magnitude between in-distribution and out-of-distribution data.
>
> Lastly, as noted in W3.2, when QPA serves as the backbone, the dataset size is only '10 × episode_length', which does not impose significant stress on the neural network. While achieving high precision in training would be ideal, considering computational costs, we only train the Morse network for an additional 200 iterations after per query. It is noteworthy that in many tasks, QPA and SURF involve training the reward model thousands of times after per query. Therefore, our use of the Morse network effectively meets our needs without incurring substantial additional computational overhead.
>
> >**Q5.** Source of the code and results deviation.
>
> >>**1).** About code.
>
> **A:** To ensure fairness in our experiments, we used the original source code provided by the authors of each baseline algorithm. Specifically, the sources are as follows:
>
> PEBBLE、SURF: *https://openreview.net/attachment?id=TfhfZLQ2EJO&name=supplementary_material*
>
> RUNE: *https://github.com/rll-research/rune*
>
> QPA: *https://github.com/huxiao09/QPA*
>
> B-Pref: *https://github.com/rll-research/BPref*
>
> The only modification we made was to unify the logging format during training. We changed QPA's logging from using wandb to the storage format used by the B-Pref framework, which is also used by PEBBLE, SURF, and RUNE.
>
> We will also include the specific sources of the code in the Appendix to provide further clarity.
>
> >>**2).** About results.
>
> **A:** Regarding the observation that "results deviate from reported literature," this may be due to differences in hardware or variations in random seeds between machines. As our research focuses on online preference-based reinforcement learning, the datasets are collected during training. We can only ensure that the results presented are those obtained from running the baseline algorithms on our machines.

---

> ### Author Response · Authors · 2024-11-15
> **Reply to Reviewer KyBu - Part 5**
>
> >**Q6.** Is there a set of reasonable hyperparameters for unseen tasks?
>
> **A:** For unseen tasks, we recommend using the hyperparameter settings for PPE provided in Appendix G.1.
>
> As discussed in Section 4.2, the key hyperparameters to consider are the parameter  $\epsilon$ required by EXT and the Mixture ratio .
>
> Regarding $\epsilon$, based on the derivation in Appendix C, it is reasonable to choose a small $\epsilon$ which ensures that $\pi_E$ and $\pi_T$ remain similar. However, setting $\epsilon$ too small may reduce exploration, so we set it to $1e-2$. The Mixture ratio controls the proportion of in-distribution versus out-of-distribution data in query selection, and a balanced ratio $0.5$ is relatively reasonable.
>
> The experiments on different tasks presented in the paper are based on these parameters, which contribute to the observed performance improvements, making them relatively reasonable and reliable. For more detailed discussions on parameter selection, please refer to Section 4.2.

---

> ### Comment · Reviewer_KyBu · 2024-11-18
>
> Thanks for your detailed response. It seems the reference to SAC was an oversight on my side. Furthermore, the proposed changes and additions will resolve most of my issues. I also deem the proposed changes viable for a potential CR version, as they are limited to the appendix or minor enough to not require a full review. Therefore, i changed my scores to reflect that.
>
> The only answer that i do not deem fully satisfactory, are the ones regarding the Morse network. You clarified everything wrt. the use within your work, but not beyond that scope. However, given the limited literature available, some general questions arise that are relevant for considering the methods potential in substantially more complex domains (like the mentioned scalability [beyond the complexity used in your domains]).

---

> > ### Author Response · Authors · 2024-11-18
> >
> > Thank you for your thoughtful review and for taking the time to reconsider your scores. We greatly appreciate your recognition of our work and are pleased that the proposed changes address most of your concerns.
> >
> > We acknowledge your feedback on the Morse network and understand the importance of exploring its potential in more complex domains. We will certainly consider this in future work.
> >
> > Thank you again for your valuable insights and support.

---

### Official Review · Reviewer_TDUG · 2024-11-04

**Soundness:** 2
**Presentation:** 2
**Contribution:** 3
**Rating:** 6
**Confidence:** 2

**Summary:**

The paper proposes the Proximal Policy Exploration (PPE) algorithm to improve the quality of reward models in preference-based reinforcement learning (PbRL) by expanding the coverage of the preference buffer used to train the reward model. To achieve this, PPE has two components: a proximal-policy extension approach to encourage exploration in out-of-distribution transitions near the current policy and a mixture distribution query method for balancing in-distribution and out-of-distribution data in the buffer. The authors evaluate PPE on DMControl and MetaWorld benchmarks.

**Strengths:**

- **Novel Approach.** The proximal-policy extension and mixture distribution query methods decompose exploration and exploitation on learning the preference model for PbRL.
- **Empirical Results.**  The experimental results show some improvement over previous methods and the authors ablate some design choices.
- **Compatibility with Existing Frameworks.** PPE is compatible with previous PbRL methods.

**Weaknesses:**

1. Line 48 states that PbRL "addresses these challenges." However, PbRL is also prone to human biases. It would be better to phrase it as "PbRL addresses some of these challenges."
2. Line 84 should read "summary" instead of "summery"
3. Figure 1(b) is missing one point for a training region of size 9.
4. While the OOD detection can be useful to encourage the agent to explore more informative (s,a) pairs, learning the function $f_\phi$ can be computationally expensive. The paper would benefit from a discussion on how training $f_\phi$ impacts wall-clock time and necessary computational resources compared to previous methods.
5. The paper does not discuss how the stopping time for feedback collection is decided. The manuscript would benefit from a further discussion on this and its impact on the training. For example, what happens in Figure 3(c) if one stops collecting feedback after 50% of training similarly to (a) and (b)? Alternatively, does RUNE outperform PPE the training is continued to $2 \times 10^6$ steps similarly to (a) and (b)?
6. Moreover, what does it mean to stop feedback collection after $1 \times 10^6$ steps in terms of number of preference queries?
7. In Algorithm 1 and 2, trajectory pairs $\tau_0, \tau_1$ are sampled from $P^{in/out}(\tau)$, however the latter is not a distribution, but rather just a real number. Could the authors clarify how the sampling is performed?
8. The paper claims that the proposed method learns better reward models because it has a larger coverage over the state-action spaces, however, there is no discussion on how the coverage differs between different algorithms. It would be interesting to have a measure of how much more of the state-action space is indeed covered.
9. Line 296 should state "the cardinality of" or "the size of" instead of "the quantity of".
10. PPE needs to maintain 4 buffers $\mathcal D, \mathcal D^{cp}, \mathcal D^p, \mathcal D^m$. This can be quite expensive in terms of memory. Moreover, what is $\mathcal D^{cp}$ exactly? How does it differ from other buffers? In Algorithm 2, line 5, in which buffer is the transition stored? What is the $\mathcal D$ buffer?
11. The plots are not properly explained. Are the authors plotting the average success rate over how many seeds? Are the shaded areas the standard deviation, standard error, or some other uncertainty measure?
Overall, the paper proposes an interesting approach to learning better reward models. However, it could benefit from more clarity in presentation, more thorough discussion of computational costs, and more detailed experimental insights.

**Questions:**

See **Weaknesses**. Additionally:
1. In line 237, if $\mathcal{A}_{uni}$ simply the action space of the MDP? It might be clearer to write $a_u \sim \mathrm{Uniform}(\mathcal A)$ instead.
2. I am not sure the Equation 5 captures the desideratum in the preceding paragraph. From my understanding, the objective of $\pi_E$ is to take actions that take the agent to unexplored areas, i.e., areas that are classified as 0 by $M_\phi$, which can be achieved by minimizing $M_\phi$, maintaining the constraints of staying close to $\pi_T$. However, Equation 5 is a maximization problem. Could the authors clarify this?
3. In lines 285 - 287, the authors state that the query selection method should actively select OOD data to increase coverage while also selecting OOD data for query. Should it instead say that it should select in-distribution data for query? If not, why?
4. During training, what it means for data to be in and out-of distribution changes. Are the elements in $\mathcal D^M$ relabeled? In line 14 and 15 of Algorithm 2, the data in $\mathcal D$ and $\mathcal D^{cp}$ is relabeled but not $\mathcal D^M$.
5. Could the authors clarify why the SAC algorithm has been chosen in Algorithm 2?
6. In line 7 of Algorithm 2, what is $\tau$?
7. What is the difference between "iteration" and "interaction" in Algorithm 2?
8. How often is $M_\phi$ updated? For how many gradient steps? How does this impact wall-clock time?
9. What are the similarities and differences of the exploration mechanism of PPE compared to the exploration induced by model-based PbRL (e.g., "HIP-RL: Hallucinated Inputs for Preference-based Reinforcement Learning in Continuous Domains" or "Efficient Preference-Based Reinforcement Learning Using Learned Dynamics Models")? Is learning an exploration policy similar to learning a model of the dynamics and exploring optimistically in the learned dynamical model?

---

> ### Author Response · Authors · 2024-11-18
> **Reply to Reviewer TDUG - Part 1**
>
> We sincerely appreciate your insightful feedback and comprehensive review. Below, we address the concerns you have raised in detail:
> >**W1,W2.** Problem of statement in line 48 and line 84
>
> **A:** Thank you for pointing out this issue. In the revised version, we will replace "addresses these challenges" with "addresses some of these challenges." Additionally, we will correct the spelling error in "summary.”
>
> >**W3.** Is Figure 1(b) missing one point for a training region of size 9?
>
> **A:** We apologize for any confusion caused by our description. However, we would like to clarify that Figure 1(b) accurately includes all points.
>
> In Section 3.1, the toy example is conducted on a grid map with both x and y coordinates ranging from 0 to 9. Therefore, in Figure 1(b), when the training region size is 9, it means the entire grid map is the training region, resulting in **no data being 'out of the training region.’**
>
> Consequently, the 'data out of training region' point is missing for a training region size of 9 in Figure 1(b) regarding the 'standard deviation of reward model.’
>
> To reduce potential misunderstandings, we will revise Line 160 from "the grid world is a 10x10 lattice" to "the grid world is structured as a 9x9 grid.”
>
> > **W4.** About  OOD detection computational  cost
>
> **A:** Thank you for your suggestion.
>
> >>**1).** Discussion on $f_\phi$
>
> Firstly, In our study, we utilized a neural network with a 3x256 architecture to learn the function $f_\phi$ required for the Morse network, as described in Eq. (4).
>
> Secondly, we do not rely on the specific outputs of the Morse network to determine whether data is OOD. Instead, we only utilize the gradient $\nabla_a M_{\phi}(s,a)$ and use it as a basis for sampling data in the 'Mixture Distribution Query'. These applications do not demand high precision in the Morse network's outputs; they only require a relative distinction in magnitude between in-distribution and out-of-distribution data.
>
> Lastly, Given that our dataset is not very large, especially when QPA is used as the backbone with a dataset size of only '10 × episode_length',  which does not impose significant stress on the neural network.
>
> Considering computational costs, we only train the Morse network for an additional 200 iterations after completing after per query. It is noteworthy that in many tasks, QPA and SURF involve training the reward model thousands of times after per query. Therefore, our use of the Morse network effectively meets our needs without incurring substantial additional computational overhead.
>
> >>**2).** Experiments results of computational cost
>
> We averaged the time required for QPA and QPA+PPE to train the reward model (and the Morse network) after five query phases on the walker_walk task, all conducted on the same machine.
>
> | Method | Average Time (seconds) |
> | --- | --- |
> | QPA | 45.29 |
> | QPA+PPE | 50.42 |
>
> While PPE does introduce additional computational overhead, training the Morse network, like the reward model, is only necessary after each query. The total number of queries varies by task. For instance, in the 'walker_walk' task, we followed the QPA setup, requiring a total of 100 preference feedbacks, with each query obtaining 10 preference feedbacks. Therefore, the overall training process does not significantly increase computational cost.
>
> We will include these detail as an additional supplement in the revised version of our paper.
>
>
> >**W5.** The paper does not discuss how the stopping time for feedback collection is decided
>
> **A:** Thank you for your insightful comments. Our experimental setup strictly adheres to the configurations outlined in the SURF and QPA papers. For instance, the training duration, query frequency, and total amount of preference feedback for the drawer_open task follow the settings specified in the SURF paper [1].
>
> The primary aim of our study is to explore whether enhancing the preference buffer's coverage of the proximal policy data distribution positively impacts practical PbRL. The current experimental results already provide evidence supporting this.
>
> Regarding your question about determining the "stopping time for feedback collection", this is indeed an intriguing topic within PbRL. We plan to explore this further in our future research.
>
> [1] *Park J, Seo Y, Shin J, et al. SURF: Semi-supervised reward learning with data augmentation for feedback-efficient preference-based reinforcement learning[J]. arXiv preprint arXiv:2203.10050, 2022.*

---

> ### Author Response · Authors · 2024-11-18
> **Reply to Reviewer TDUG - Part 2**
>
> >**W6.** What does it mean to stop feedback collection after $1\times10^6$ steps in terms of the number of preference queries?
>
> **A:** Based on my understanding, stopping feedback collection after 1e6 steps implies that no additional preference feedback will be gathered, and the reward model will remain unchanged thereafter.
>
> In our study, we aligned our experimental setup with those of prior works (PEBBLE, SURF, RUNE, QPA) to effectively assess the efficacy of our method. As such, we did not extensively investigate the impact of variations in experimental settings on performance.
>
> >**W7.** Could the authors clarify how the sampling is performed?
>
> **A:** In Equation (8), we define $P^{in/out}(\tau)$ as the probability density function for the behavior segment $\tau$. In Algorithms 1 and 2, the notation '$\tau^0, \tau^1 \sim P^{in/out}(\tau)$' indicates that the trajectory pairs are sampled from the probability distribution described by $P^{in/out}(\tau)$. This mathematical notation is similar to that used in Line 9 of Algorithm 1 in PER[1].
>
> Specifically, when PPE uses QPA as the backbone, $\mathcal{D}^{cp}$ stores the most recent '10 * episode_length' tuples of $(s, a, r, s')$ data. During the preference feedback query phase, we select two continuous segments $\tau^0, \tau^1$ of length segment_length (typically 50, as commonly used in SURF, PEBBLE, QPA, and RUNE experiments) to query the preference label $y_p$, thereby forming the preference pair $(\tau^0, \tau^1, y_p)$ mentioned in Section 2.
>
> In other words, $\mathcal{D}^{cp}$ contains '10 * (episode_length - segment_length + 1)' segments of $\tau$ data. In our work, we construct a probability density function $P^{in/out}(\tau)$ to describe the probability distribution over $\tau$. We then perform sampling operations from this described distribution.
>
> [1] *Schaul T. Prioritized Experience Replay[J]. arXiv preprint arXiv:1511.05952, 2015.*
>
> >**W8.** It would be interesting to have a measure of how much more of the state-action space is indeed covered
>
> **A:** Thank you for your insightful suggestion. It is indeed an interesting idea to measure the coverage of the state-action space. However, it's important to note that in practical PbRL tasks, the dimensions of the state and action spaces are typically quite high, making it challenging to intuitively measure and display coverage. Therefore, in Section 3.1 of our paper , we took a different approach by using a low-dimensional toy example to demonstrate the impact of coverage on the reward model.
>
> Moreover, existing methods do not typically optimize for the coverage of the preference buffer. Our approach explicitly targets this as an optimization goal, which undoubtedly enhances the coverage of the preference buffer.
>
> >**W9.** Statement of line296
>
> **A:** Thank you for pointing this out. We will replace "the quantity of" with "the size of" in the revised version.

---

> ### Author Response · Authors · 2024-11-18
> **Reply to Reviewer TDUG - Part 3**
>
> >**W10.** About 4 buffers, $\mathcal{D},\mathcal{D}^{cp},\mathcal{D}^{p},\mathcal{D}^{m}$
>
> **A:** Thank you for your questions. Let me first provide an overview of the components in a typical PbRL setup:
>
> a).data collection  →
>
> b).data selection and preference labeling  →
>
> c).learning the reward model using preference labels $(\tau^0,\tau^1,y_p)$ →
>
> d).optimizing $\pi_T$ with the learned reward model via reinforcement learning methods.
>
> >>**1).** What is $\mathcal{D}^{cp}$？
>
> As explained in lines 290-292 of the paper, $\mathcal{D}^{cp}$ stores potential segments $\tau$ that might be selected during the "data selection and preference labeling" phase. Specifically, when selecting $(\tau^0, \tau^1)$ for preference labeling, these segments are drawn from $\mathcal{D}^{cp}$.
>
> In the code [1], this corresponds to the 'inputs' parameter of the 'RewardModel' class, which has a 'max_size' limiting the number of **episodic trajectories** stored. For instance, PEBBLE [2] sets 'max_size' to 100, while QPA [3] sets it to 10. Instead of storing all possible $\tau$, it stores past episodic trajectories like $(s_0, a_0, r_0, s_1, \ldots, s_T)$, where 'T' denotes the terminal state.
>
> >>**2).** How does it differ from other buffers? What is $\mathcal{D}$ buffer?
>
> - **$\mathcal{D}$**: This is the replay buffer, a fundamental concept in reinforcement learning, storing $(s, a, \hat{r}, s')$ instead of the ground truth $r$. It is used during the policy optimization phase with the learned reward model.
> - **$\mathcal{D}^{p}$**: In the code [1], this corresponds to 'buffer_seg1', 'buffer_seg2', and 'buffer_label' in the 'RewardModel' class. It stores preference signals $(\tau^0, \tau^1, y_p)$ for learning the reward model.
> - **$\mathcal{D}^m$**: This buffer stores an additional one-dimensional data $M_\phi(s, a)$ for each $(s, a)$ in $\mathcal{D}^{cp}$, as shown in Equation (7). It is used to compute $M_\phi(\tau)$ to assess the OOD degree of $\tau$.
>
> >> **3).** Is it quite expensive in terms of memory?
>
> - $\mathcal{D}$ is essential for all off-policy reinforcement learning algorithms as a replay buffer.
> - $\mathcal{D}^{cp}$ and $\mathcal{D}^{p}$ are necessary for existing online PbRL methods.
> - $\mathcal{D}^m$ only requires storing an additional one-dimensional value $M_\phi(s, a)$ for each $(s, a)$ in $\mathcal{D}^{cp}$, which is a minor addition performed in Algorithm 2, line 4.
>
> Therefore, PPE does not require significantly more memory compared to previous online PbRL methods.
>
> >>**4).** In Algorithm 2, line 5, in which buffer is the transition stored?
>
> The newly collected $(s, a)$ transitions are stored in both $\mathcal{D}$ and $\mathcal{D}^{cp}$.
>
> [1]*B-Pref: https://github.com/rll-research/BPref*
>
> [2]*PEBBLE、SURF:https://openreview.net/attachment？id=TfhfZLQ2EJO&name=supplementary_material*
>
> [3]*QPA:https://github.com/huxiao09/QPA*
>
> >**W11.** The plots are not properly explained
>
> **A:** Thank you for your valuable feedback. We appreciate your suggestions for improving the clarity of our presentation.
>
> In our experiments, we used five different seeds to compute the average performance. The shaded areas in the plots represent the 95% confidence intervals.
>
> We will include this information in the appendix of the revised version to ensure clarity and transparency.

---

> ### Author Response · Authors · 2024-11-18
> **Reply to Reviewer TDUG - Part 4**
>
> >**Q1.**  A clearer formulation of of $\mathcal{A}_{uni}$，in line 237
>
> **A:** Thank you for your suggestion. We will provide the clearer formulation $a_u\sim \text{Uniform}(\mathcal{A})$ in line 237 in the revised version of the paper.
>
> >**Q2.**  Not sure the Equation 5 captures the desideratum in the preceding paragraph.
>
> **A:** Thank you very much for your insightful feedback. Your understanding is indeed correct. We made an error in Equation (3), which should be revised to:
>
> $ M_\phi(s,a) = 1 - K_{RBF}(f_\phi(s,a),a), \text{ where } K_{RBF}(z_1,z_2) = e^{-\frac{\lambda^2}{2}\|z_1-z_2\|^2}. $
>
> We intend for $M_\phi$ to output values approaching 1 for OOD data and values approaching 0 for in-distribution data.
> However, this correction does not affect the description or derivation related to Equation (5).
>
> We will address this in the revised version of the paper.
>
> >**Q3.** Question In lines 285 - 287
>
> **A:** Thank you for your observation. You are absolutely correct.
>
> This was indeed a presentation error on our part. In the revised version of the paper, we will correct line 287 by replacing "out-of-distribution" with "in-distribution."
>
> >**Q4.** Are the elements in $\mathcal{D}^{m}$ relabeled?
>
> **A:** Yes, in Algorithm 2, line 15, the elements in $\mathcal{D}^m$ are indeed relabeled.
>
> As explained in W10.1 and W10.2, $\mathcal{D}^{cp}$ does not require relabeling because it stores ground truth data.
>
> Furthermore, $\mathcal{D}^m$ does not require a separate storage space for $(s, a, M_\phi(s, a))$. Instead, $\mathcal{D}^m$ simply maintains an additional one-dimensional data $M_\phi(s, a)$ for each $(s, a)$ in $\mathcal{D}^{cp}$.
>
> Therefore, relabeling $\mathcal{D}^m$ involves "*Relabeling the OOD metric via $M_\phi(\cdot)$ for $(s, a) \in \mathcal{D}^{cp}$*",in Algorithm 2, line 15.
>
> >**Q5.** Why  SAC ?
>
> **A:** Thank you for your question.
>
> We chose the SAC algorithm to ensure consistency and fairness in our comparisons. SAC is used for policy optimization in several related works, including PEBBLE, RUNE, SURF, and QPA.
>
> By using the same algorithm, we can more accurately assess the effectiveness of our proposed method while controlling for variables across different experiments.
>
> >**Q6.** What is $\tau$?
>
> **A:** As mentioned in Section 3.2, line 224, $\tau$ refers to a sequence of state-action pairs, specifically $\{(s_t, a_t), \ldots, (s_{t+H}, a_{t+H})\}$. This sequence represents a trajectory segment used in the context of the algorithm.
>
> >**Q7.** Difference between "iteration" and "interaction" in Algorithm 2
>
> **A:** Thank you for your observation. In the context of Algorithm 2, there is no significant difference between "iteration" and "interaction."
>
> However, to avoid any potential confusion, we will standardize the terminology and use "iteration" consistently throughout the revised version of the paper.
>
> >**Q8.** Details about $M_\phi$
>
> >>**1).** How often is $M_\phi$ updated?
>
> **A:** $M_\phi$ is updated after each query, aligning with the update frequency of the reward model.
>
> This ensures that $M_\phi$ remains consistent with the most recent data and model adjustments.
>
> >>**2).** For how many gradient steps?
>
> **A:** We update $M_\phi$ for 200 gradient steps after each query.
>
> In contrast, existing methods [1,2] generally update the reward model with over a thousand gradient steps per query.
>
> Consequently, from an implementation perspective, we believe PPE does not introduce significant additional computational overhead.
>
> [1] *Park, Jongjin, et al. "SURF: Semi-supervised reward learning with data augmentation for feedback-efficient preference-based reinforcement learning." arXiv preprint arXiv:2203.10050 (2022).*
>
> [2] *Hu X, Li J, Zhan X, et al. Query-policy misalignment in preference-based reinforcement learning[J]. arXiv preprint arXiv:2305.17400, 2023.*
>
> >>**3).** How does this impact wall-clock time?
>
> **A:** We have addressed the problem about impact on wall-clock time in **W4**.

---

> ### Author Response · Authors · 2024-11-18
> **Reply to Reviewer TDUG - Part 5**
>
> >**Q9.** What are the similarities and differences of the exploration mechanism of PPE compared to the exploration induced by model-based PbRL? Is learning an exploration policy similar to learning a model of the dynamics and exploring optimistically in the learned dynamical model?
>
> **A:** To clarify the exploration mechanisms, let's break down PbRL into the following stages:
>
> a).data collection  →
>
> b).data selection and preference labeling  →
>
> c).learning the reward model using preference labels $(\tau^0,\tau^1,y_p)$ →
>
> d).optimizing $\pi_T$ with the learned reward model via reinforcement learning methods .
>
> The primary contribution of PPE lies in stage **a**, where it focuses on collecting more data that is near-policy and out-of-preference buffer distribution. This enhances the coverage of the data used to train the reward model, leading to a more reliable evaluation of the current policy.
>
> >>**1).** The similarities and differences
>
> **Compare to [1]**
>
> The exploration mechanism in [1] (**"Efficient Preference-Based Reinforcement Learning Using Learned Dynamics Models"**) aims to improve PbRL efficiency using partial human demonstrations and large amounts of unlabeled (s, a, s') data. This approach primarily contributes to stages **c** and **d** by augmenting trajectories with a model-based approach and using an MPC-like strategy for policy selection. The exploration in [1] is designed to learn a more accurate state transition model (Algorithm 1, line 2).
>
> - **Similarities:** Both approaches aim to enhance coverage to improve model accuracy.
>
> - **Differences:**
> [1] seeks to improve global coverage of (*s*,*a*), whereas PPE focuses only on enhancing coverage within the near-policy distribution. The purposes, problems addressed, and settings of exploration in PPE and [1] differ significantly.
>
> **Compare to [2]**
>
> [2] ("HIP-RL: Hallucinated Inputs for Preference-based Reinforcement Learning in Continuous Domains") uses learned state transition models and reward models to explore optimal strategies over a future horizon, acknowledging the potential inaccuracies in these models.
>
> - **Differences:** The motivation of PPE differs from [2]. PPE aims to develop a more reliable reward model, while [2] seeks to maximize a learned fake reward $\hat{r}$ during the Policy Search phase (Algorithm 1, line 5), which is conducted **offline**  as noted in Section 3.
>
>
>
> >>**2).** Is learning an exploration policy similar to learning a model of the dynamics and exploring optimistically in the learned dynamical model?
>
> This is indeed an intriguing question that warrants thoughtful consideration.
>
> In my view, exploration serves as a means to an end, and different methods employ this means to address distinct problems.
>
> While there may be scenarios where they appear similar, I believe the approaches and motivations of [1] and [2] differ significantly from those of PPE.
>
> In [1] and [2], exploration is used to learn a more accurate dynamics model and optimize $\pi_T$ within that model. This approach focuses on leveraging exploration to build and utilize a reliable dynamics model. The problem they address is "how to construct and apply a more reliable dynamics model."
>
> On the other hand, PPE uses exploration to address the problem of "how to construct a more reliable reward model for evaluating the current policy."
>
> Therefore, the motivations behind these approaches are not similar.
>
> [1] *Liu, Yi, et al. "Efficient preference-based reinforcement learning using learned dynamics models." *2023 IEEE International Conference on Robotics and Automation (ICRA)*. IEEE, 2023.*
>
> [2] *Zhang, Chen Bo Calvin, and Giorgia Ramponi. "HIP-RL: Hallucinated Inputs for Preference-based Reinforcement Learning in Continuous Domains." *ICML 2023 Workshop The Many Facets of Preference-Based Learning*. 2023.*

---

> ### Author Response · Authors · 2024-11-23
> **Looking Forward to Further Discussion**
>
> Dear Reviewer TDUG,
>
> We sincerely appreciate your valuable time and effort in reviewing our paper. We hope that our responses have adequately addressed all your questions and concerns.
>
> If there are any remaining issues or if you require further information, please do not hesitate to let us know. We are more than happy to provide additional materials or discuss any aspect of our work in more detail.
>
> Thank you once again for your consideration.
>
> Best regards,
>
> The Authors

---

> ### Comment · Reviewer_TDUG · 2024-11-26
>
> We thank the authors for the clarifications. I still have some questions and concerns detailed below.
>
> > In Equation (8), we define ...
>
> To confirm my understanding: given the trajectories in $\mathcal D_{cp}$​, do you calculate the probability of each trajectory being in- or out-of-distribution and then sample one accordingly?
>
> > However, it's important to note that in practical PbRL tasks, the dimensions of the state and action spaces are typically quite high, making it challenging to intuitively measure and display coverage. [...] Moreover, existing methods do not typically optimize for the coverage of the preference buffer. Our approach explicitly targets this as an optimization goal, which undoubtedly enhances the coverage of the preference buffer.
>
> I agree that in practical PbRL settings, the state-action space is often high-dimensional, making coverage assessment non-trivial. In [1], the authors addressed this challenge by discretizing the state-action space to visualize coverage. Given that this method explicitly optimizes for preference buffer coverage, it is critical to demonstrate that this optimization directly contributes to PPE’s effectiveness. While this is a plausible explanation, I am not fully convinced that the observed behavior is necessarily emergent from this objective alone.
>
> This point is particularly significant because the authors highlight in their [response to Reviewer pvxk](https://openreview.net/forum?id=gXV84CnMUm&noteId=nl7b20cq96). that "Our primary contribution focuses on addressing the problem of insufficient coverage in the preference buffer, which can lead to unreliable evaluations by the reward model". Without clearer evidence linking the optimization goal to the claimed benefits, the argument remains somewhat speculative.
>
> > As explained in lines 290-292 of the paper, $\mathcal D_{cp}$ stores potential segments $\tau$ that might be selected during the "data selection and preference labeling" phase. Specifically, when selecting $(\tau_0, \tau_{1})$ for preference labeling, these segments are drawn from $\mathcal D_{cp}$.
>
> Just to make sure I understand this correctly. Is $\mathcal D_{cp}$ a queue with maximum size 100?
>
> Additionally, while I am familiar with the concept of a replay buffer, the fact that the method maintains four separate buffers could benefit from greater clarity. Explicitly outlining the purpose of each buffer would greatly aid the reader's understanding.
>
> Finally, thank you for clarifying where the transitions are stored. It would be helpful to make this explicit in the algorithm itself to ensure the process is clear to readers.
>
> > In our experiments, we used five different seeds to compute the average performance. The shaded areas in the plots represent the 95% confidence intervals.
>
> I appreciate the clarification.
>
> > As mentioned in Section 3.2, line 224 ...
>
> I understand the general definition of a trajectory. However, I am unclear about the origin of the specific trajectory $\tau$ for which $P^{in}(\tau)$ is computed. Is my understanding correct that, to construct the sampling distribution, all probabilities are calculated for the available trajectories at each iteration, and then one is sampled accordingly?
>
> Additionally, I noticed that in Equation (6), $\tau$ is used both as the input trajectory and in the denominator. To avoid potential confusion, it might be helpful to use different notation for the denominator, such as $\tau'$.
>
> ---
>
> The authors mention that the Morse network serves as a tool for OOD detection and can be replaced by other OOD metrics, such as ensemble networks or RND.
>
> Would these alternative methods perform worse than the Morse network? What is the specific reason for choosing Morse networks over these other methods? Clarifying this comparison could help justify the choice of the Morse network over other potential techniques. It would also strengthen the paper to include experimental results using other OOD detection mechanisms to show that PPE can be applied to any detection method.
>
> ---
>
> Additionally, I noticed that the authors only present experimental results for Hammer, Sweep-Into and Drawer-Open and not other tasks considered in SURF. Is there any specific reason for the chosen tasks? The paper could benefit from further experimental evaluation to show that it consistently outperforms prior methods.
>
> ---
>
> Lastly, I believe the paper could benefit from improvements in its presentation as it still is fairly hard to follow.
>
> **References**
>
> [1] Hazan, Elad, et al. "Provably efficient maximum entropy exploration." _International Conference on Machine Learning_. PMLR, 2019.

---

> > ### Author Response · Authors · 2024-11-29
> > **Discussion with Reviewer TDUG - Part 2**
> >
> > > **5.** Would these alternative methods perform worse than the Morse network? What is the specific reason for choosing Morse networks over these other methods? Clarifying this comparison could help justify the choice of the Morse network over other potential techniques. It would also strengthen the paper to include experimental results using other OOD detection mechanisms to show that PPE can be applied to any detection method.
> >
> > >>**1).** Why Morse network?
> >
> > **A:** Thank you for your suggestion. Our core investigation focuses on whether increasing the preference buffer coverage can optimize the performance of online PbRL.
> >
> > This is supported by the motivation example in Section 3.1, the COVERAGE VISUALIZATION in Appendix H, and extensive experiments throughout the paper.
> >
> > The Morse network in our work is only used to provide gradients to $\pi_E$ for exploring OOD data.
> >
> > Compared to ensemble-based OOD detection mechanisms like RND, the Morse network offers several advantages:
> >
> > 1. The Morse network requires only a single neural network, whereas RND requires multiple networks for OOD detection, thus reducing computational overhead.
> > 2. The output of the Morse network is within the range [0,1]. In contrast, the output of RND is derived from the difference between the neural network's output and a random network, and this output is not standardized.
> >
> > Specifically, the form of the Morse network used in this paper is:
> >
> > $
> > \min\limits_\phi \frac{1}{N} \sum_{s,a\sim \mathcal{D}^p} \left[\frac{\lambda^2}{2} ||f_\phi(s,a)-a||^2 + \frac{1}{M} \sum_{a_u\sim \text{Uniform}(\mathcal{A})} \exp^{-\frac{\lambda^2}{2}||f_\phi(s,a_u)-a_u||^2}\right]
> > $
> >
> > If we were to use the RND form, it would be:
> >
> > $
> > \min\limits_\phi \frac{1}{N} \sum_{s,a\sim \mathcal{D}^p} \left[||f_\phi(s,a)-f_{\text{random}}(s,a)||^2\right]
> > $
> >
> > By setting the RND's random network to $f_{\text{random}}(s,a) = a$, it matches the left part of the optimization objective of the Morse network used in our paper.
> >
> > Therefore, the Morse network has the additional objective of "moving away from out-of-distribution (s,a)" and requires fewer neural networks compared to RND.
> >
> > >> **2).** It would also strengthen the paper to include experimental results using other OOD detection mechanisms to show that PPE can be applied to any detection method.
> >
> > **A:**  Thank you for your suggestion. We have added a set of experimental results using RND, as shown below.
> >
> > | Task | Morse Network | RND |
> > | --- | --- | --- |
> > | drawer-open | 69.81 ± 43.41 | **79.52 ± 44.53** |
> > | Sweep-into | **96.47 ± 8.47** | 77.17 ± 20.69 |
> > | Hammer | **96.27 ± 5.19** | 77.41 ± 16.71 |
> >
> >
> > > **6.** The paper could benefit from further experimental evaluation to show that it consistently outperforms prior methods.
> >
> > **A:** Thank you for your suggestion. We have conducted additional experiments on other Meta-World tasks, and the results are as follows:
> >
> > | Task         | PEBBLE         | SURF           | RUNE           | QPA            | PPE            |
> > |--------------|----------------|----------------|----------------|----------------|----------------|
> > | door-open    | 72.4 ± 35.87   | 91.4 ± 11.04   | 98.2 ± 1.92    | 100.0 ± 0.00   | **100.0 ± 0.00** |
> > | door-unlock  | 99.6 ± 0.55    | 99.6 ± 0.55    | **100.0 ± 0.00** | 98.4 ± 1.14    | **100.0 ± 0.00** |
> > | button-press | 76.6 ± 6.02    | 78.8 ± 14.38   | 88.6 ± 7.13    | 72.2 ± 29.81   | **90.0 ± 5.43**  |
> > | window-open  | 87.1 ± 17.08   | 87.2 ± 26.99   | 96.7 ± 4.14    | 90.1 ± 20.09   | **100.0 ± 0.00** |
> >
> > *I look forward to further discussions with you to present this work in the best possible way to our readers.*

---

> > > ### Comment · Reviewer_TDUG · 2024-12-03
> > >
> > > We thank the authors for their detailed and thoughtful response. Given the efforts put into addressing the concerns raised and the engaging discussion during the rebuttal period, I have decided to increase the score to a 6. However, I maintain a low confidence in my evaluation (2).
> > >
> > > I concur with Reviewer 7kxy’s point regarding the need for improved clarity in presentation, which could enhance the overall accessibility of the work.
> > >
> > > Additionally, I would like to express some reservations regarding the coverage results. While the authors have provided some evidence, I am not yet fully convinced by the results, as these are based on a single task with a single seed. The results would be more compelling if additional experiments, covering multiple tasks and seeds, were included to strengthen the claims. Additionally, I found the definition of coverage in Figure 6 unclear. Does it measure the number of s,a-clusters visited? If so, it seems inconsistent with the range of values observed in Figure 5, which appears to be [1, 5]. Providing a precise explanation of the metric would greatly aid in interpreting the results.

---

> > > > ### Author Response · Authors · 2024-12-03
> > > > **Discussion with Reviewer TDUG - Part 3**
> > > >
> > > > We greatly appreciate your recognition of our paper and response, and thank you for raising the score.
> > > >
> > > > > **Clarity.**
> > > >
> > > > **A:** In the initial feedback, the reviewer pointed out some areas in our presentation that could cause confusion, including suggestions for phrasing improvements, a detailed explanation of the Motivating Example, details about the Morse network, and clarifications on the meanings of various buffers. After thorough discussions with the reviewer, these issues have been addressed more clearly in the revised version. We will also carefully review the manuscript to further eliminate any potential ambiguities.
> > > >
> > > > > **Not yet fully convinced by the results, as these are based on a single task with a single seed.**
> > > >
> > > > **A:** We completely agree that using more seeds is essential to fully demonstrate performance advantages. We would like to clarify that Figures 5 and 6 report results based on a single seed due to the short submission window for the revised version. We have now included results using five different fixed seed values, which are consistent with the conclusions already reported. We will incorporate these findings into future versions of the manuscript.
> > > >
> > > > > **I found the definition of coverage in Figure 6 unclear. Does it measure the number of s,a-clusters visited?**
> > > >
> > > > **A:** Yes, indeed.
> > > > In Figure 6, we record the number of s,a-clusters visited (an s,a-cluster is considered covered if the number of corresponding (s,a) pairs is ≥1, with a total of 200 s,a-clusters, calculated as 10$\times$20).
> > > >
> > > > To better illustrate the coverage, we have also included a heatmap showing the coverage levels across different s,a-clusters. In Figure 5, we use a heatmap to record the number of (s,a) pairs corresponding to each s,a-cluster, capping the maximum value at 5 to more clearly display the differences in the number of (s,a) pairs across different s,a-clusters.

---

> ### Author Response · Authors · 2024-11-29
> **Discussion with Reviewer TDUG - Part 1**
>
> Thank you very much for your response. We appreciate your valuable suggestions for improving the quality of our paper. We have provided explanations below to address the concerns you still have. We hope to continue our discussion further.
>
> > **1.** Do you calculate the probability of each trajectory being in- or out-of-distribution and then sample one accordingly?
>
> **A:** You are  correct. The specific implementation is as follows:
>
> ```python
> with torch.no_grad():
>     # Calculate M_\phi(tau) using M_\phi(s,a)
>     metrics = torch.stack([torch.FloatTensor(temp).to(self.device) for temp in self.ood_metrics[:max_len]], dim=0)
>     ood_seg = metrics.unfold(1, self.size_segment, 1)
>     prob = ood_seg.mean(-1).flatten()
>     # Sample from P^{out}
>     sample_indices_flat1 = torch.multinomial(prob, mb_size, replacement=True)
>     sample_indices_flat2 = torch.multinomial(prob, mb_size, replacement=True)
>
> ```
>
> Similarly, to sample $\tau$ from the in-distribution, you only need to replace 'self.ood_metrics' with '1-self.ood_metrics' in the above code.
>
> Additionally, $\mathcal{D}^{cp}$ contains '10 * (episode_length - segment_length + 1)' segments of $\tau$ data.
>
> > **2.** Clearer evidence linking the optimization goal to the claimed benefits
>
> **A:** Thank you for your suggestion.
>
> In the revised version, we have added a more intuitive visualization of coverage in Appendix H (COVERAGE VISUALIZATION).
>
> Specifically, we saved the preference buffer for both QPA+PPE and QPA under the same random seed in the Walker_walk task. We then used K-means to discretize the state and action spaces into 10 and 20 cluster centers, respectively.
>
> We recorded the coverage of the 10x20 grid by the preference buffer for different methods under varying numbers of feedback as follows:
> |  | QPA | QPA+PPE |
> | --- | --- | --- |
> | Feedback=10 | 111 | 111 |
> | Feedback=20 | 145 | 150 |
> | Feedback=30 | 158 | 169 |
> | Feedback=40 | 165 | 178 |
> | Feedback=50 | 172 | 192 |
> | Feedback=60 | 177 | 196 |
> | Feedback=70 | 187 | 197 |
> | Feedback=80 | 191 | 197 |
> | Feedback=90 | 192 | 198 |
> | Feedback=100 | 194 | 198 |
>
> The table shows the area covered by the discretized 10x20 grid. It is evident that the QPA+PPE method, which actively explores out-of-distribution  areas, indeed enhances the coverage of the preference buffer compared to QPA.
>
> Combined with the conclusion from Section 3.1 that "the broader the preference buffer coverage, the more reliable the reward model's evaluation capability," we can conclude that PPE actively expands the preference buffer coverage, thereby enabling the use of a more reliable reward model for policy improvement.
>
> > **3.** About Buffers
>
> >>**1).** Is $\mathcal{D}^{cp}$  a queue with maximum size 100?
>
> **A:** The size of $\mathcal{D}^{cp}$ depends on the parameter settings of the backbone algorithm. PPE itself does not impose any restrictions. Specifically, when using QPA+PPE, the size of $\mathcal{D}^{cp}$ is 10; when using PEBBLE+PPE, the size of $\mathcal{D}^{cp}$ is 100.
>
> >> **2).** Clarity
>
> **A:** Thank you for your feedback. We have added additional explanations in Appendix I.2, which we believe will help reduce any potential misunderstandings for the readers.
>
> > **4.** About $\tau$
>
> >> **1).** Is my understanding correct that, to construct the sampling distribution, all probabilities are calculated for the available trajectories at each iteration, and then one is sampled accordingly?
>
> **A:** Your understanding is mostly correct. However, it is important to clarify that sampling is **not required at each iteration**. Sampling is only needed during the reward learning phase, which occurs after each query is completed.
>
> Specifically, in the Walker_walk task, the feedback limit is 100, and each query can obtain 10 feedbacks. Therefore, only 10 queries are needed in this task. Consequently, reward learning also occurs only 10 times, meaning the operation of calculating the sampling distribution is performed just 10 times throughout the entire training process. This does not significantly increase computational overhead. We have also provided related experimental explanations in response to your question W4.2.
>
> >> **2).** To avoid potential confusion, it might be helpful to use different notation for the denominator, such as $\tau’$.
>
> **A:** Thank you very much for your suggestion. We have made the changes in the revised version to reduce any potential misunderstandings.

---

### Official Review · Reviewer_pvxk · 2024-11-04

**Soundness:** 3
**Presentation:** 2
**Contribution:** 2
**Rating:** 5
**Confidence:** 4

**Summary:**

To address the issue of significant human involvement in preference-based reinforcement learning (PbRL), this work proposes the Proximal Policy Exploration (PPE) method, which is designed to purposefully explore and extend the coverage of the preference buffer, thereby enhancing the evaluation ability of the reward model and the subsequent value function in PbRL. This improvement aims to promote an unbiased transmission of human intentions to the agent's behavior. Specifically, this work first employs a Morse Neural Network to identify the distributional properties of interactive transition samples, jointly constructing a policy regularization objective that encourages the agent to explore targeted transitions. Finally, PPE introduces the mixture distribution query to balance the inclusion of both in-distribution and out-of-distribution data. The experiments demonstrate that PPE achieves significant improvements in both human feedback efficiency and sample efficiency.

**Strengths:**

This work addresses the accuracy of the reward model in preference-based reinforcement learning from a different perspective—by expanding the coverage of the preference buffer, which is a compelling and meaningful direction. The work specifically designs the proximal policy extension and mixture distribution query methods to explore policy behaviors outside the preference distribution but near the policy distribution, enabling the agent to effectively leverage both the existing preference distribution data and the unexplored state-action space beyond the preference buffer. Experimentally, compared to the state-of-the-art method QPA, PPE achieves significant performance improvements across DMC tasks of varying difficulty levels and includes extensive ablation studies on components and key parameters. Overall, the method in this work is highly reproducible, well-developed, and provides theoretical guarantees for certain conclusions, with a comprehensive experimental setup.

**Weaknesses:**

The overall originality and novelty of the specific methodology in this paper are relatively limited. The problem addressed here essentially relates to a common issue of OOD detection and balanced utilization. The distinguishing factor is its application to the preference-based reinforcement learning (PbRL) setting, specifically aiming to address inaccuracies in the learned reward model due to insufficient coverage in the preference buffer. Unfortunately, the methods adopted do not introduce particularly novel approaches to solve this challenge.
In detail, i）The paper applies a morse network to assess the distribution of each transition; however, this approach was previously introduced in "Offline Reinforcement Learning with Behavioral Supervisor Tuning."；ii）Secondly, the authors incorporate this network within a regularized policy objective, which encourages the agent to purposefully explore targeted transitions. However, the initial objective form is heavily based on prior work. The main contribution here is the proposal to tighten the constraints, enabling a more effective closed-form approximation of the original objective. iii）Lastly, the authors introduce a Mixture Distribution Query method that queries both in-distribution and out-of-distribution data, which is the main but limited contribution of this work.

Experimentally, the performance on the MetaWorld tasks is suboptimal, failing to achieve consistent performance improvements. Also, there are some deficiencies in the writing of this paper. For example, there is excessive verbosity, and some problem statements are not clearly expressed.

**Questions:**

1) In Section 3.1, some descriptions lack clarity. For instance, in Figure 1b, it is unclear why the variance of the reward model can be used to identify whether a transition sample belongs to the training region. Therefore, it is essential to provide additional explain or experimental evidence demonstrating the discriminative ability of the reward model variance with respect to the training data distribution before using the conclusion.

2) The mixture distribution query process focuses on enhancing the coverage of the preference buffer and improving the evaluation capability of the reward model by combining in-distribution and out-of-distribution data. However, it remains unclear how the preference labels for newly explored trajectory pairs {τ0, τ1, y}^b_i=1 are generated. Does this process still require human involvement? This aspect is not addressed in the paper. If similarity metrics-based automatic labeling is used, there may be issues with accuracy, while if human intervention is still needed, this approach does not effectively solve the initial problem.

3) There is a potential but possibly flawed doubt here. In the mixture distribution query method, if there is a computationally intensive query operation, the prior policy regularization setup may seem less essential, as the agent would still have the capability to explore targeted transitions without any policy regularization. Distributed query on such naive policy exploration can also achieve similar effects. Hence, why is a mixture distribution query still necessary? Can similar functions or effects be achieved by coordinating the previous parameter λ or other possible adjustments?

4) Some descriptions and causal relationships in the manuscript are unclear. For example, in lines 180-181, "Therefore, the method proposed by Liang et al. (2022)...", what method has been proposed in the existing work? If so, it is difficult for us to understand the subsequent results based on the previous sentence. Finally, there are some repeated descriptions, such as lines 243-247, 275-278, etc., which can be appropriately simplified to express accurately and concisely.

5) Although the problem solved in this paper falls within the scope of PbRL, PbRL-related preliminaries are rarely involved in the method section, so this part can be simplified. In terms of formal description, the manuscript can focus more on the problem itself.

---

> ### Author Response · Authors · 2024-11-19
> **Reply to Reviewer pvxk - Part 1**
>
> We deeply appreciate your insightful feedback and thorough review. Please see our detailed responses to the concerns you've highlighted below:
>
> >**W1.** The paper applies a morse network to assess the distribution of each transition; however, this approach was previously introduced in "Offline Reinforcement Learning with Behavioral Supervisor Tuning.”
>
> **A:** We fully acknowledge that our use of the Morse network for  OOD detection was inspired by the approach outlined in [1]. However, we respectfully clarify that this is merely a tool for OOD detection and represents just one aspect of the practical implementation in our paper.
>
> Our primary contribution focuses on addressing the problem of insufficient coverage in the preference buffer, which can lead to unreliable evaluations by the reward model.
>
> Section 3.1 uses a motivating example to demonstrate this problem.
> This issue is critical because only when the reward model's evaluation capability is reliable can the direction of policy optimization be correct.
> To tackle this, we propose the PPE algorithm, which enhances the coverage of the preference buffer concerning the proximal policy's data distribution. This improvement increases the reliability of the reward model's evaluation of the current policy, thereby optimizing the performance of online PbRL.
>
> In essence, **the main contribution of PPE is identifying and addressing the fundamental issue revealed in Section 3.1**. The Morse network is simply a tool that can be replaced by other OOD metrics, such as ensemble networks or RND [2].
>
> [1] *Srinivasan P, Knottenbelt W. Offline Reinforcement Learning with Behavioral Supervisor Tuning[J]. arXiv preprint arXiv:2404.16399, 2024.*
>
> [2]  *Burda, Yuri, et al. "Exploration by random network distillation." arXiv preprint arXiv:1810.12894 (2018).*
>
> >**W2.** Secondly, the authors incorporate this network within a regularized policy objective, which encourages the agent to purposefully explore targeted transitions. However, the initial objective form is heavily based on prior work.
>
> **A:** The Morse network in [1] addresses the issue of overly strict constraints on OOD actions in offline settings, implementing a dynamic adjustment.
>
> While both [1] and PPE formalize their problems as constrained optimization problems, this is a classic mathematical approach in optimization theory. **The key difference lies in how each paper addresses these constraints and the distinct meanings behind them.**
>
> In [1], the KL constraint aims to align the target policy $\pi$ with the distribution of offline data $\pi_\beta$. Here, $\pi_\beta$ is not directly accessible, and $\pi$ is the neural network being optimized.
> In contrast, PPE seeks to align the exploration policy $\pi_E$ with the target policy $\pi_T$. The target policy $\pi_T$ is optimized through SAC in other stages, and $\pi_E$ can be approximated in closed form using this mathematical framework.
>
> PPE emphasizes exploring OOD data near the policy to increase the coverage of the preference buffer distribution, necessitating high **efficiency** in data collection. This is why PPE presents a closed-form approximation.
> In [1], the setting is offline, meaning exploration is not applicable. Due to the nature of offline settings, [1] can further expand on $\pi_\beta$ in Section 4.2 [1], which is infeasible in an online setting where no stable dataset exists.
>
> Therefore, aside from formalizing the problems as constrained optimization issues, the motivations, methods for solving these problems, and their implications are entirely different.
> **Both papers describe distinct mathematical problems in different domains using similar mathematical forms and solve them through different approaches.**
>
> [1] *Srinivasan P, Knottenbelt W. Offline Reinforcement Learning with Behavioral Supervisor Tuning[J]. arXiv preprint arXiv:2404.16399, 2024.*
>
> >**W3.** Lastly, the authors introduce a Mixture Distribution Query method that queries both in-distribution and out-of-distribution data, which is the main but limited contribution of this work.
>
> **A:** In Section 3.1, we use a motivating example to discuss the impact of reduced feedback within the same training region on the reward model, as illustrated in Figure 1.d and lines 182-187.
>
> Focusing solely on exploring OOD data without adequately increasing the amount of in-distribution data can compromise the reliability of the reward model in those regions. This is further validated by the ablation study in Section 4.2, as shown in Figure 4.a.
>
> These findings demonstrate that the Mixture Distribution Query is essential for the proximal-policy extension, and both are crucial for the performance improvement of the PPE algorithm.

---

> ### Author Response · Authors · 2024-11-19
> **Reply to Reviewer pvxk - Part 2**
>
> >**W4.** Experimentally, the performance on the MetaWorld tasks is suboptimal, failing to achieve consistent performance improvements.
>
> **A:** In the context of online PbRL algorithms with limited preference feedback, our method demonstrates a noticeable improvement over baseline algorithms on the MetaWorld tasks. Detailed data supporting this can be found in Appendix D, Table 1.
>
> It is important to note that online PbRL does not have direct access to ground truth rewards like traditional RL algorithms. Therefore, achieving better performance with limited preference feedback is inherently challenging and remains a key focus in the development of the online PbRL field.
>
> >**W5.** Also, there are some deficiencies in the writing of this paper. For example, there is excessive verbosity, and some problem statements are not clearly expressed.
>
> **A:** Thank you for your feedback. We will incorporate the reviewers' comments to present a clearer and more concise version in the revised manuscript.

---

> ### Author Response · Authors · 2024-11-19
> **Reply to Reviewer pvxk - Part 3**
>
> >**Q1.** In Section 3.1, some descriptions lack clarity. For instance, in Figure 1b, it is unclear why the variance of the reward model can be used to identify whether a transition sample belongs to the training region. Therefore, it is essential to provide additional explain or experimental evidence demonstrating the discriminative ability of the reward model variance with respect to the training data distribution before using the conclusion.
>
> **A:** Thank you for your feedback. We apologize for any confusion caused by the writing and are happy to clarify. It seems there may have been a misunderstanding regarding the conclusion drawn from Figure 1.b.
>
> In Section 3.1, lines 179-182, we state: "*Figure 1.b demonstrates that the variance in outputs from ensemble reward models, given the same transition input, **does not enable** distinction of whether the transition belongs to the training region. Therefore, the method proposed by Liang et al. (2022) cannot actively expand the preference buffer’s coverage.*"
>
>
> As highlighted in bold, our intended meaning is actually the opposite of what you might have understood. We are emphasizing that the variance of the reward model cannot be used to identify whether a transition sample belongs to the training region.
>
> >**Q2.** The mixture distribution query process focuses on enhancing the coverage of the preference buffer and improving the evaluation capability of the reward model by combining in-distribution and out-of-distribution data. However, it remains unclear how the preference labels for newly explored trajectory pairs $ ( \tau^0, \tau^1, y_p )^b_{i=1}$ are generated. Does this process still require human involvement? This aspect is not addressed in the paper. If similarity metrics-based automatic labeling is used, there may be issues with accuracy, while if human intervention is still needed, this approach does not effectively solve the initial problem.
>
> **A:** Thank you for your question. I would like to clarify this point, and these clarifications will also be included in the revised manuscript.
>
> Let me first provide an overview of the components in a typical PbRL setup:
>
> a) Data collection →
>
> b) Data selection and preference labeling →
>
> c) Learning the reward model using preference labels $(\tau^0, \tau^1, y_p)$ →
>
> d) Optimizing $\pi_T$ with the learned reward model via reinforcement learning methods
>
> In stage **b**, algorithms typically select $(\tau^0, \tau^1)$ pairs, which are then submitted for human preference labeling. In most PbRL implementations, scripts are typically used to simulate human preference labeling [1, 2, 3]. Our paper follows the same setup.
>
> The Mixture Distribution Query is used only in stage **b** to select $(\tau^0, \tau^1)_{i=1}^b$, as shown in Algorithm 1. These selected pairs are then submitted for human preference labeling (Algorithm 2, line 8). **This is the only stage that requires human involvement.**
>
> This process is consistent with what is described in PEBBLE (Algorithm 2, line 11), QPA [5] (Algorithm 1, line 6), and RUNE (Algorithm 1, line 9).
>
> [1] *B-Pref: https://github.com/rll-research/BPref*
>
> [2] *PEBBLE、SURF:https://openreview.net/attachment？id=TfhfZLQ2EJO&name=supplementary_material*
>
> [3] *QPA:https://github.com/huxiao09/QPA*
>
> [4] *Lee K, Smith L, Abbeel P. Pebble: Feedback-efficient interactive reinforcement learning via relabeling experience and unsupervised pre-training[J]. arXiv preprint arXiv:2106.05091, 2021.*
>
> [5] *Hu X, Li J, Zhan X, et al. Query-policy misalignment in preference-based reinforcement learning[J]. arXiv preprint arXiv:2305.17400, 2023.*
>
> [6] *Liang X, Shu K, Lee K, et al. Reward uncertainty for exploration in preference-based reinforcement learning[J]. arXiv preprint arXiv:2205.12401, 2022.*

---

> ### Author Response · Authors · 2024-11-19
> **Reply to Reviewer pvxk - Part 4**
>
> >**Q3.**  There is a potential but possibly flawed doubt here. In the mixture distribution query method, if there is a computationally intensive query operation, the prior policy regularization setup may seem less essential, as the agent would still have the capability to explore targeted transitions without any policy regularization. Distributed query on such naive policy exploration can also achieve similar effects. Hence, why is a mixture distribution query still necessary? Can similar functions or effects be achieved by coordinating the previous parameter λ or other possible adjustments?
>
> **A:** Perhaps your concern stems from the relationship between proximal-policy extension (EXT) and mixture distribution query (MDQ). If so, let me clarify:
>
> **EXT**: Through the closed-form approximation in Eq(6), EXT promotes the exploration of **new OOD (s, a) pairs**.
>
> **MDQ**: As described in Algorithm 1, MDQ samples $(\tau^0, \tau^1)$ from existing (s, a) pairs. Note that $\tau = (s_t, a_t, \ldots, s_{t+H}, a_{t+H})$.
>
> 1. **Intuitive Analysis**
> - Using only MDQ without EXT means there is no active data collection phase. This approach merely maximizes the use of the existing $\mathcal{D}^{cp}$. Without EXT, the coverage of the proximal policy's data distribution in $\mathcal{D}^{cp}$ would not increase directly. Thus, regardless of the query selection method, the preference buffer's distribution coverage would inevitably be less than when using EXT, as EXT actively collects OOD data. Solely relying on MDQ contradicts the motivation of our paper.
> - Using only EXT without MDQ is discussed in W3.
> 2. **Experimental Analysis**
> - In Section 4.2, Figure 4.a, we discuss the necessity of both MDQ and EXT for the algorithm. The experimental results demonstrate that EXT and MDQ complement each other. EXT increases the coverage of the candidate data buffer $\mathcal{D}^{cp}$, while MDQ efficiently extracts $(\tau^0, \tau^1)$ with high OOD metrics from $\mathcal{D}^{cp}$.
>
> In my opinion, the issues addressed by EXT and MDQ cannot be resolved by merely adjusting the parameter λ. This approach is currently the most straightforward solution we have devised. If you have any suggestions, we would be delighted to discuss them further.
>
> >**Q4.** Some descriptions and causal relationships in the manuscript are unclear. For example, in lines 180-181, "Therefore, the method proposed by Liang et al. (2022)...", what method has been proposed in the existing work? If so, it is difficult for us to understand the subsequent results based on the previous sentence. Finally, there are some repeated descriptions, such as lines 243-247, 275-278, etc., which can be appropriately simplified to express accurately and concisely.
>
> >>**1).** in lines 180-181, "Therefore, the method proposed by Liang et al. (2022)...", what method has been proposed in the existing work?
>
> **A:** Thank you for your feedback. We will replace "Therefore, the method proposed by Liang et al. (2022)..." with "Therefore, RUNE, proposed by Liang et al. (2022)...".
>
> The purpose of this statement is to use a motivating example, shown in Figure 1.b, to illustrate that the method of reward uncertainty used by RUNE is unable to distinguish OOD data effectively.
>
> >>**2).** Finally, there are some repeated descriptions, such as lines 243-247, 275-278, etc., which can be appropriately simplified to express accurately and concisely.
>
> **A:** Thank you for your suggestion. In the revised version, we will simplify the expressions in lines 243-247 and 275-278 for greater accuracy and conciseness.
>
> >**Q5.**  Although the problem solved in this paper falls within the scope of PbRL, PbRL-related preliminaries are rarely involved in the method section, so this part can be simplified. In terms of formal description, the manuscript can focus more on the problem itself.
>
> **A:** Thank you for your insightful feedback. In the revised version, we will consider relocating some of the PbRL-related preliminary content to the Appendix.

---

> ### Author Response · Authors · 2024-11-23
> **Looking Forward to Further Discussion**
>
> Dear Reviewer pvxk,
>
> We sincerely appreciate your valuable time and effort in reviewing our paper. We hope that our responses have adequately addressed all your questions and concerns.
>
> If there are any remaining issues or if you require further information, please do not hesitate to let us know. We are more than happy to provide additional materials or discuss any aspect of our work in more detail.
>
> Thank you once again for your consideration.
>
> Best regards,
>
> The Authors

---

> > ### Comment · Reviewer_pvxk · 2024-11-28
> >
> > Thanks for the detailed responses. Most of my concerns have been addressed. However, I still think two of the three main parts of this paper draw upon existing work significantly, which limits the novelty of the work. Experimentally, the evaluation of MetaWorld tasks with high variance lacks sufficient comprehensiveness, and the observed improvements are also limited. The original rating is unchanged.

---

> > > ### Author Response · Authors · 2024-12-01
> > > **Discussion with Reviewer pvxk - Part 1**
> > >
> > > Thank you for your response. We are very pleased that our reply can address most of your concerns. For the remaining questions you have, we provide the following clarifications:
> > >
> > > > **About novelty**
> > >
> > > >>**From Motivation**
> > >
> > > **A:** The core question of this paper is, "Does increasing the preference buffer's coverage of nearby policies improve the performance of online PbRL?"
> > >
> > > Undoubtedly, through the toy example in Section 3.1, the COVERAGE VISUALIZATION in Appendix H, and the extensive experimental data provided in the paper, we reveal that increasing the preference coverage of the near policy distribution can indeed further optimize online PbRL.
> > >
> > > According to our research, no one in the online PbRL community has explored a similar topic before. We have demonstrated through a series of experiments a topic that has not been studied, which I believe is innovative.
> > >
> > > >>  **From Implementation**
> > >
> > > **A:** We utilized the OOD detection capability of the Morse Network to realize our motivation, and the object of OOD measurement is the preference buffer. The purpose is to collect OOD data for the preference buffer to improve the quality of the reward model,  which has not been attempted in prior studies.
> > >
> > > We used a closed-form approximation and provided related proofs, which is a more efficient form of data collection compared to intrinsic rewards and better suits our needs. This is also designed entirely for the need for efficient data collection in online PbRL, which is also innovative.
> > >
> > > Additionally, based on the experimental phenomena in Section 3.1, we designed the Mixture Distribution Query and proximal-policy extension to complement each other, which is also unprecedented work.
> > >
> > > Overall, apart from using existing OOD detection methods, the process from *Problem Motivation* -> *Revealing Phenomena* -> *Problem Modeling* -> *Handling Constrained Optimization Problems* all contain innovative aspects.
> > >
> > > The ultimate problem this paper needs to address is, "Does increasing the preference buffer's coverage of nearby policies improve the performance of online PbRL?" Spending effort to design a new and potentially ineffective OOD detection method itself may not have much significance here.
> > >
> > > **Therefore, overall, we believe this paper does possess a certain degree of innovation, and it is indeed a new exploration in the online PbRL community.**

---

> > > ### Author Response · Authors · 2024-12-01
> > > **Discussion with Reviewer pvxk - Part 2**
> > >
> > > > **About performance**
> > >
> > > >>**High variance**
> > >
> > > **A:** **In QPA [1] Figure 6, SURF [2] Figure 2, PEBBLE [3] Figure 4, and RUNE [4] Figure 3, the performance of these prior PbRL algorithms in the MetaWorld environment shows high variance.** Even the performance of training SAC with ground truth rewards, as provided in [1, 2, 3, 4], exhibits significant variance.
> > >
> > > We attribute this phenomenon to the fact that in MetaWorld, performance is measured by success rate, unlike in DMControl tasks where cumulative reward is used as the performance metric. This issue does not appear to be specific to our method but rather a common occurrence in online PbRL experiments on MetaWorld tasks.
> > >
> > >
> > > Furthermore, we have discovered an interesting phenomenon that can also support this viewpoint. In Metaworld tasks, the optimization goal is to maximize the sum of Hand-engineered Reward provided by the Metaworld tasks. However, the final evaluation metric is the Success Rate. Sometimes, an agent with a lower sum of Hand-engineered Reward may actually achieve a higher Success Rate.
> > >
> > > For example, in the Window-open environment, as shown in the table below:
> > >
> > >
> > > |  | Success Rate  | Hand-engineered Reward |
> > > |-----------|------------------|------------------------|
> > > | PEBBLE    | 87.1       ± 17.08              | 2280.85        ± 1206.06|
> > > | PPE       | **100.0     ± 0.00**               | **3176.80         ± 1048.38**   |
> > > | QPA       | 90.1      ± 20.09              | 2919.79     ± 890.34      |
> > > | RUNE      | 96.7      ± 4.14               | 1721.57      ±842.82      |
> > > | SURF      | 87.2      ± 26.99             | 2633.09         ±1292.82   |
> > >
> > >
> > > The final performance of RUNE corresponds to a Hand-engineered Return that is lower than all other algorithms, yet its policy achieves a success rate second only to PPE. This, to some extent, illustrates the issue of inconsistency between the optimization objective and the evaluation metric.
> > >
> > > Therefore, we additionally provide a table using Hand-engineered Reward as the evaluation standard in Metaworld tasks to better demonstrate the performance improvements brought by our method.
> > >
> > > | Task | PEBBLE | PPE | QPA | RUNE | SURF |
> > > | --- | --- | --- | --- | --- | --- |
> > > | hammer | 2968.92 ± 1730.12 | **4560.82 ± 110.70** | 3971.48 ± 1249.63 | 4403.38 ± 397.31 | 4069.37 ± 482.93 |
> > > | swep-into | 3673.08 ± 2032.38 | **4301.48 ± 422.62** | 3149.74 ± 1309.21 | 3817.07 ± 1204.22 | 2989.84 ± 1940.13 |
> > > | drawer-open | 3062.54 ± 928.66 | **3890.57 ± 772.18** | 3452.50 ± 1137.78 | 3715.66 ± 971.34 | 3421.41 ± 1071.75 |
> > > | door-open | 3205.08 ± 1322.95 | **4339.99 ± 99.36** | 4271.51 ± 240.36 | 3240.79 ± 1026.21 | 3293.31 ± 963.42 |
> > > | door-unlock | 4180.98 ± 153.27 | 4156.87 ± 311.68 | **4194.23 ± 434.18** | 3968.38 ± 621.11 | 4199.83 ± 357.92 |
> > > | button-press | 3178.47 ± 70.02 | **3302.26 ± 90.94** | 2821.22 ± 269.36 | 3121.58 ± 109.43 | 3214.49 ± 60.57 |
> > >
> > > Overall, the PPE algorithm does indeed bring performance improvements.
> > >
> > > [1]*Hu X, Li J, Zhan X, et al. Query-policy misalignment in preference-based reinforcement learning[J]. arXiv preprint arXiv:2305.17400, 2023.*
> > >
> > > [2]*Park J, Seo Y, Shin J, et al. SURF: Semi-supervised reward learning with data augmentation for feedback-efficient preference-based reinforcement learning[J]. arXiv preprint arXiv:2203.10050, 2022.*
> > >
> > > [3]*Lee K, Smith L, Abbeel P. Pebble: Feedback-efficient interactive reinforcement learning via relabeling experience and unsupervised pre-training[J]. arXiv preprint arXiv:2106.05091, 2021.*
> > >
> > > [4]*Liang X, Shu K, Lee K, et al. Reward uncertainty for exploration in preference-based reinforcement learning[J]. arXiv preprint arXiv:2205.12401, 2022.*

---

> > > ### Author Response · Authors · 2024-12-01
> > > **Discussion with Reviewer pvxk - Part3**
> > >
> > > >>improvements
> > >
> > > To more clearly demonstrate the impact of PPE on performance, we also conducted statistical analysis using the Welch's T-test, as mentioned in [1], on all experimental data shown in this paper. We compared the final performance and all evaluations during training between PPE and each baseline algorithm across all tasks. A positive value indicates the degree of improvement of PPE over the corresponding baseline method.
> > >
> > > The results of the Welsh T-test comparing the final performance of PPE with that of other baselines：
> > >
> > > |  | PEBBLE | SURF | RUNE | QPA |
> > > | --- | --- | --- | --- | --- |
> > > | walker_walk | 6.0247 | 5.1586 | 5.7963 | 1.5858 |
> > > | Walker-run | 5.7399 | 4.0779 | 4.1103 | 0.9492 |
> > > | Quadruped-walk | 2.9398 | 1.1491 | 1.4278 | 0.6062 |
> > > | Quadruped-run | 1.5780 | 2.1105 | 3.4360 |  0.6763 |
> > > | Cheetah-run | 1.6605 | 2.9569 | 1.6168 | 1.0810 |
> > > | Humanoid-stand | 3.4696 | 12.0554 | 2.5250 | 6.7143 |
> > > | Drawer-open | 1.7871 | 0.9496 | 0.7384 | 0.9496 |
> > > | Sweep-into | 1.3052 | 2.4492 | -0.8298 | 1.2485 |
> > > | Hammer | 2.2834 | 0.9249 | 0.5327 | 0.8834 |
> > >
> > > The results of the Welsh T-test comparing the performance of PPE with that of other baselines across all evaluations during training:
> > >
> > > |  | PEBBLE | SURF | RUNE | QPA |
> > > | --- | --- | --- | --- | --- |
> > > | walker_walk | 17.2363 | 7.7909 | 18.6282 | 4.4137 |
> > > | Walker-run | 31.8804 | 23.5488 | 26.6891 | 5.9516 |
> > > | Quadruped-walk | 14.4218 | 6.5593 | 11.7086 | 1.0420 |
> > > | Quadruped-run | 13.1904 | 16.8964 | 20.4076 |  4.5345 |
> > > | Cheetah-run | 11.5897 | 6.9901 | 8.7536 | 5.8251 |
> > > | Humanoid-stand | 26.5792 | 24.8984 | 23.5906 | 7.9869 |
> > > | Drawer-open | 10.4972 | 2.0464 | -1.3821 | 2.6691 |
> > > | Sweep-into | 7.0637 | 1.1292 | -5.8169 | 2.1000 |
> > > | Hammer | 16.4993 | 8.7253 | 1.0280 | 2.4130 |
> > >
> > > **Overall, as shown in Figure 2, Figure 3, and Table 1, the PPE algorithm provides a noticeable performance improvement over previous work.**
> > >
> > > [1] *https://arxiv.org/abs/1904.06979*
> > >
> > > *We hope our further clarifications can effectively address your concerns, and we look forward to further communication and discussion.*

---

> > > ### Author Response · Authors · 2024-12-02
> > > **Discussion with Reviewer pvxk - Part ４**
> > >
> > > >> **p-values**
> > >
> > > We have also compiled the p-values for the Welch's T-test comparing the performance of PPE with other baselines across all evaluations during training in the table below:
> > >
> > > | **Task/Algorithm** | **PEBBLE** | **SURF** | **RUNE** | **QPA** |
> > > | --- | --- | --- | --- | --- |
> > > | Walker_walk | <0.001 | <0.001 | <0.001 | <0.001 |
> > > | Walker_run | <0.001 | <0.001 | <0.001 | <0.001 |
> > > | Quadruped-walk | <0.001 | <0.001 | <0.001 | 0.2977 |
> > > | Quadruped-run | <0.001 | <0.001 | <0.001 | <0.001 |
> > > | Cheetah-run | <0.001 | <0.001 | <0.001 | <0.001 |
> > > | Humanoid-stand | <0.001 | <0.001 | <0.001 | <0.001 |
> > > | Drawer-open | <0.001 | 0.0412 | 0.1675 | 0.0078 |
> > > | Hammer | <0.001 | <0.001 | 0.3041 | 0.0160 |
> > > | Swep-into | <0.001 | <0.001 | 0.0184 | <0.001 |
> > >
> > > For most tasks, the algorithms PEBBLE, SURF, RUNE, and QPA show significant differences from PPE (p-value < 0.05), indicating that these algorithms perform significantly differently from PPE on these tasks.
> > >
> > > In three cases, the p-value is greater than 0.05, such as for RUNE in the Drawer-open task, RUNE in the Hammer task, and QPA in the Quadruped-walk task. Although there is no significant difference, PPE remains one of the best methods.

---

> > > ### Author Response · Authors · 2024-12-04
> > > **Discussion with Reviewer pvxk - Part 5**
> > >
> > > Thank you for your efforts in reviewing our work. In response to the concerns you raised, we have provided explanations and experimental evidence in '**Discussion with Reviewer pvxk - Part1, Part2, Part3, Part4**'. We hope these clarifications address your questions. If you find them satisfactory and consider adjusting the score, we would be extremely grateful.

---

### Author Response · Authors · 2024-11-21
**Response to Reviewer Feedback and Invitation for Further Discussion**

Dear Reviewers,

Thank you for your valuable feedback. We have revised the manuscript based on your suggestions, and the updated version has been uploaded.

All changes in the revised manuscript are highlighted in purple. The main text required only minor modifications, while additional experiments and explanations have been added to the Appendix.

The key revisions are as follows:

- **[pvxk, TDUG, KyBu, 7kxy]:** We addressed minor suggestions, including corrections for spelling, structural adjustments, and expression optimizations for clarity and precision.

- **[TDUG, 7kxy]:** In Section 4, we added details about the experimental seeds and provided explanations for the error bars in the plots of experimental results.

- **[pvxk]:**  We moved some PbRL-related preliminaries to Appendix A.

- **[KyBu, 7kxy]:**  In Appendix E, we included comparative experiments for PEBBLE+PPE, PEBBLE+RUNE, and PEBBLE, as well as QPA+PPE, QPA+RUNE, and QPA, to demonstrate PPE's compatibility and advantages over current exploration methods.

- **[TDUG, KyBu]:**  Appendix F now includes detailed implementation details of the Morse network and an analysis of computational costs.

- **[TDUG, KyBu]:**  Appendix G provides additional explanations regarding the motivating example.

- **[pvxk, TDUG, KyBu, 7kxy]:**  Appendix H provides a comprehensive overview of the PbRL pipeline, including baseline code sources and implementation details.

**We sincerely hope to engage in further discussions with you regarding this work and to address any remaining questions you may have.**

---

> ### Author Response · Authors · 2024-12-02
> **Response to Reviewer Feedback and Invitation for Further Discussion - Part 2**
>
> Based on the reviewers' latest round of feedback, we have conducted the following additional experiments:
>
> -----
>
> > **[TDUG].** COVERAGE VISUALIZATION
>
> We have added COVERAGE VISUALIZATION (already updated in Appendix H) to more intuitively demonstrate the effect of PPE on expanding the coverage of the preference buffer.
>
> >**[7kxy]**. Human feedback experiments
>
> Human feedback experiments (already updated in Appendix I.6) have been conducted to demonstrate the feasibility of the PPE method in real-world human feedback scenarios. These experiments show that by introducing exploration of out-of-distribution data near the policy, PPE enhances the diversity of candidate $\tau \in \mathcal{D}^{cp}$, thereby mitigating the issue of overly similar behavioral segments being presented by QPA to humans for preference labeling during the query phase.
>
> To more clearly observe the differences between agents trained with QPA+PPE and QPA under human feedback settings, we have uploaded videos of the final agent performance and the query videos provided to humans by different algorithms to the following **anonymous link**:
>
> [https://anonymous.4open.science/r/Demos-ICLRDiscussion](https://anonymous.4open.science/r/Demos-ICLRDiscussion/query_ppe2.mp4)
>
> >**[pvxk, 7kxy].** Welch's T-test
>
> We have applied Welch's T-test to all experimental results. The t-statistics and p-values clearly demonstrate the performance improvements brought by the PPE method.
>
> > **[TDUG].**  Verify the feasibility of using alternative OOD detection methods
>
>  We conducted experiments replacing the Morse network in PPE with RND-based OOD detection to verify the feasibility of using alternative OOD detection methods in place of the Morse network and to highlight the advantages of using the Morse network.
>
> > **[TDUG].** Additional experiments on Metaworld tasks
>
> Additional experiments on Metaworld tasks, including door-open, door-unlock, button-press, and window-open, have been conducted to further validate the effectiveness of the PPE method across diverse environments.
>
> ----
> *We sincerely **thank all the reviewers for their valuable time and effort in reviewing our paper**. We hope our responses have adequately addressed all questions and concerns.*
>
> *Should there be any remaining issues or a need for further information, please do not hesitate to contact us. **We are more than willing to provide additional materials or discuss any aspect of our work in greater detail.***

---

> > ### Author Response · Authors · 2024-12-04
> > **Response to Reviewer Feedback - Part 3**
> >
> > We sincerely appreciate the hard work of all the reviewers during the discussion phase. We firmly believe that these discussions have significantly enhanced the quality of the paper.
> >
> > During the discussion phase, we engaged in open and in-depth exchanges with the reviewers and reached a consensus on many key issues, successfully addressing most of the concerns. These improvements will be incorporated into future versions of the manuscript, with specific updates and explanations summarized in Part 1 and Part 2.
> >
> > We will continue to address the current limitations of this work in future research. Once again, we thank all the reviewers for their valuable suggestions and support.

---

### Meta-Review · Area_Chair_9w6N · 2024-12-21

**Metareview:**

The paper studied the Proximal Policy Exploration (PPE) algorithm. The algorithm aims to improve the quality of the reward model in preference-based setting by expanding the coverage of the preference buffer. More specifically, it uses a proximal-policy extension approach to encourage exploration near the current policy and a mixture distribution query method to balance in and out of distribution data in the buffer. The authors evaluate PPE on DMControl and MetaWorld benchmarks. The weaknesses of the work are in its novelty compared to prior work, limited improvement in experiments, and clarity of the results.

**Additional Comments On Reviewer Discussion:**

The reviewers all acknowledged the authors' effort during the AC reviewer discussion period. The paper is at the borderline. The remaining concerns of the papers are about its novelty and experiments. One reviewer is still concerned about the novelty of the work when compared to prior works and its limited experimental improvement in the world. One reviewer is still concerned about the clarity of the results and expressed concern about reproducing these statistical test results by themselves. The reviewer also strongly encouraged the authors to include more human data. We encourage the authors to carefully include all the additional experiment results and the statistical tests into the future versions of the paper.

---

### Decision · Program_Chairs · 2025-01-22

Reject